# Bandits with Mean Bounds

**Nihal Sharma**                                                  *nihal.sharma@utexas.edu*
*The University of Texas at Austin*

**Soumya Basu**                                                   *basusoumya@google.com*
*Google*

**Karthikeyan Shanmugam**                                         *karthikeyanvs@google.com*
*Google DeepMind*

**Sanjay Shakkottai**                                             *sanjay.shakkottai@utexas.edu*
*The University of Texas at Austin*

**Reviewed on OpenReview:** *https://openreview.net/forum?id=4TZ4DE24fX*

## Abstract

We study a variant of the bandit problem where side information in the form of bounds on the mean of each arm is provided. We prove that these translate to tighter estimates of subgaussian factors and develop novel algorithms that exploit these estimates. In the linear setting, we present the Restricted-set OFUL (R-OFUL) algorithm that additionally uses the geometric properties of the problem to (potentially) restrict the set of arms being played and reduce exploration rates for suboptimal arms. In the stochastic case, we propose the non-optimistic Global Under-Explore (GLUE) algorithm which employs the inferred subgaussian estimates to adapt the rate of exploration for the arms. We analyze the regret of R-OFUL and GLUE, showing that our regret upper bounds are never worse than that of the standard OFUL and UCB algorithms respectively. Further, we also consider a practically motivated setting of learning from confounded logs where mean bounds appear naturally.

## 1 Introduction

We study the problem of bandits with mean bounds and bounded rewards where an agent is presented with a set of $K$ arms, along with side-information in the form of upper and lower bounds on the average reward for each of the arms. The agent is then asked to successively choose arms based on previous observations in order to maximize the cumulative reward. As is standard in MABs, the agent's performance is compared to that of a genie that always chooses the arm that gives the largest reward in expectation. We consider this in the commonly studied frameworks of stochastic Multi-Armed Bandits (MABs) and the Linear Bandit. In the former, rewards of each arm are drawn independently from its associated distribution and are uncorrelated with rewards of other arms. The linear setting couples the rewards from all arms through an unknown, but fixed latent vector that is used to paramterize the mean rewards. We seek to design arm selection policies that efficiently utilize the provided mean bounds in offering both improve regret performance and computational complexity.

Our setting is motivated by the problem of inferring efficacy of interventions from confounded logs. As a concrete example, consider the following healthcare example: Interavenous tissue plasminogen (tPA) activators are known to be highly effective in treating acute ischemic strokes if administered within 3 hours of the onset of symptoms. Otherwise, a less-effective medical therapy is recommended, as tPA causes higher chances of adverse side effects and hemorrhages Powers et al. (2019). If a log is then generated without recording the time since the onset of symptoms, it would present tPA to be better than the alternative. Naively using such a log to infer tPA to be the best intervention could lead to unfavourable outcomes in areas with poor access

to healthcare, where patients take longer to reach the hospital after symptoms appear. Inferring the optimal decision in the presence partial contextual information is non-trivial in general, however, one can extract bounds on the mean of the treatments using these logs.

The offline problem of extracting bounds on mean effects of interventions is well-studied by works such as Robins et al. (2000); Brumback et al. (2004); Richardson et al. (2014); Zhang & Bareinboim (2017); Yadlowsky et al. (2022). We are interested in using these bounds to aid the online learning of optimal actions, also studied in Zhang & Bareinboim (2017); Combes et al. (2017) for the Bernoulli MABs (the latter also treats gaussian rewards). In this work, we show that using the mean bounds in non-trivial ways can lead to improvements in regret performance over existing methods in the case of linear bandits and stochastic MABs with this side information.

The key intuition for our improvements comes from the notion of subgaussian factors of bounded random variables. A random variable taking values in $[0, 1]$ is known to be $1/2-$subgaussian, which is tight for a Bernoulli random variable with mean $\frac{1}{2}$. Given additional information about the location of its mean (in an interval, say), tighter estimates of the variance can be inferred. However, whether such information provides a sharper subgaussian factor, which in itself is an upper bound to the variance, is not known. The worst-case factor above is commonly used in the bandit setting, to characterize the concentration behavior of (bounded) random variables around their mean. Further, the jump from estimation to decision-making means that estimates can be incorrect (e.g., biased, poor confidence) so long as it does not alter the decision. In a bandit setting, this observation has been classically used to explore sub-optimal actions at a lower rate than that for the best action; this leads to a worse accuracy bound for the reward estimates of sub-optimal actions but does not affect decision-making. In this paper, we go beyond this intuition: we show that, surprisingly, known bounds on mean rewards for some actions (side information) enable us to explore other *possibly unrelated actions* at lower-rates, thus improving overall cumulative regret. Our contributions are detailed below:

**Contribution 1: Improved Subgaussian Factors with Mean Bounds:** We provide a characterization of the subgaussian factor $\sigma \leq \frac{1}{2}$ of any random variable bounded in $[0, 1]$ when upper and lower bounds on its mean are known. Specifically, in Theorem 3 and Corollary 3.1, we show that when the mean is known to be towards either half of this interval, one can infer factors that are strictly less than 0.5. These immediately imply tighter concentration bounds for bounded random variables. This result could be of independent interest, however, we study the effects of such information in bandit learning.

**Contribution 2: Linear Bandits:** We present the Restricted-set Optimism in the Face of Uncertainty for Linear bandits (R-OFUL) algorithm for bounded rewards. R-OFUL first uses the structure imposed by the linear rewards to refine the given side information and produces the tightest possible mean bounds for each arm in the action set before online interaction. Then, at each time, it leverages the geometry of the problem to restrict the set of arms to be considered. Finally, it reduces exploration by using the sharp estimates of subgaussian factors above for arms in the restricted set and chooses actions much like the standard OFUL algorithm of Abbasi-Yadkori et al. (2011).

We show that the lower bounds are key to our improvements in the online phase of the algorithm. First, the restricted set is constructed as a cone around the arm with the *largest lower bound*. Combining the lower bounds of the arms that remain with the boundedness of rewards then leads to our reduced exploration rates. Our analysis in Theorem 4 shows that using side information, R-OFUL can improve over the regret guarantees of standard OFUL by a constant factor. It also improves the computational cost due to the restriction of arms. To the best of our knowledge, this is the first investigation on Linear Bandits with Mean bounds.

**Contribution 3: Stochastic MABs:** We develop GLobal Under-Explore (GLUE)—an index based policy which, unlike the UCB Auer et al. (2002) and kl-UCB Cappé et al. (2013) algorithms, is not optimistic; i.e. the indices do not serve as high-probability upper bounds to true means of each arm. We use the fact that violating the upper bound property for the indices of suboptimal arms does not adversely affect the regret as long as the property is maintained for the (unknown) best arm. In particular, our indices for an arm are formed using a quantity that can be strictly lesser than the true subgaussian factor of the arm *only if* the arm is sub-optimal. This causes the sub-optimal arms to be under-explored which leads to improvement in regret performance.

This problem is a specific form of the Structured Bandit framework of Combes et al. (2017) where the authors develop OSSB, a non-optimistic algorithm to balance exploration across arms in structured spaces. In the case of Bernoulli rewards, OSSB reduces to the B-kl-UCB algorithm of Zhang & Bareinboim (2017) that uses the kl-UCB index for each arm truncated at the corresponding *upper bound*. In both B-kl-UCB and OSSB, the lower bounds on arm means are only used initially to prune away arms that can be identified as suboptimal, and the upper bounds are used to clip the arm indices at each time.

For general bounded rewards, the upper bound on s.g-factor of the optimal arm can be inferred from the *highest lower bound* of arm means. Therefore, we define the exploration rates of the arms to be the minimum of the individual arm subgaussian factors and the aforementioned upper bound. Our analysis in Theorem 6 shows that in instances with non-informative mean bounds, we recover the performance of UCB. However, using our adapted rates in instances with rich side information leads to GLUE significant improvements over vanilla UCB. Empirically, we see that our performance is comparable to B-KL-UCB when it is known that one of the arms has a mean close to 1.

**Contribution 4: Mean bounds from confounded logs:** We develop techniques to extract upper and lower bounds on the means of arm rewards from partially confounded logged data. Specifically, we consider a dataset that has been collected by an oracle that observes the full context, takes the optimal action and receives the corresponding rewards. However, the log only contains some parts of the context along with the corresponding (action, reward) pair, generalizing the work of Zhang & Bareinboim (2017), where none of the contexts are recorded. In the stochastic case, using bounds on the gap between the means of the best and second best arms *as observed by the oracle*, as well as the corresponding gap between best and worst arms, we derive upper and lower bounds on the mean rewards of arms to be used by an agent that acts only based *only* on the recorded parts of the context. We show that these are tight, i.e. there are instances that meet both the upper and lower bounds. We also show that these bounds can be inferred in a linear setting without the knowledge of gaps. We validate our work through synthetic and semi-synthetic experiments with the Movielens 1M dataset Harper & Konstan (2015).

## 1.1 Related Work

Bandit problems have seen a lot of interest over the past few decades (see Bubeck et al. (2012); Lattimore & Szepesvári (2020) for comprehensive surveys). A vein of generalization for the same has seen numerous advances in incorporating several forms of side information to induce further structure into this setting. Notable among them are graph-structures information (Buccapatnam et al., 2014; Valko et al., 2014; Amin et al., 2012), latent models (Li et al., 2010; Bareinboim et al., 2015; Lattimore et al., 2016; Sen et al., 2017), expert models (Auer et al., 1995; Mannor & Shamir, 2011), smoothness of the search space (Kleinberg et al., 2008; Srinivas et al., 2010; Bubeck et al., 2011), among several others. We assume side information in the form of mean bounds and study how such information affects the decision making process.

In the stochastic setting, our bandit problem has connections to the works by Zhang & Bareinboim (2017) and Combes et al. (2017). An in-depth comparison with these can be found in Section 5.3. Along another thread, Bubeck et al. (2013) provide algorithms with bounded regret if the mean of the best arm and a lower bound on the minimum suboptimality gap is known. These techniques, however, do not apply in our setting as the side information we consider does not allow us to extract such quantities in general. In the linear setting, with time-varying action sets, the works of Li et al. (2010); Abbasi-Yadkori et al. (2011) are inspired by the upper confidence bound-type arguments of Auer et al. (2002) for the stochastic case. When action sets remain fixed over time, arm-elimination type algorithms like ones in Valko et al. (2014) improve dependence on the dimension of the arms. We study the novel setting of linear bandits with mean bounds and varying action sets in this work.

The extraction of mean bounds from confounded logs has been studied in the context of estimating treatment effects in the presence of confounders. Here, actions are treatments, and the rewards capture the effects of this choice. A line of existing work performs sensitivity analysis by varying a model on the latents, measured variables, treatments and outcomes in a way that is consistent with the observed data (Robins et al., 2000; Brumback et al., 2004; Richardson et al., 2014). Recently in (Yadlowsky et al., 2022; Zhao et al., 2019), a universal bound on the ratio of selection bias due to the unobserved confounder is assumed. This means that

the treatment choice has a bounded sensitivity on the unknown context (i.e., mostly irrelevant). We deal with the other extreme, where we assume that the outcomes in the log are recorded using an unknown *optimal policy* (under complete information), and that the knowledge of worst case sub-optimality gaps for the given latent context space is known. Our assumption allows for strong dependence on the unknown context.

The use of logged data to improve online learning has been studied recently by (Zhang & Bareinboim, 2017; Zhang et al., 2019; Ye et al., 2020). The first assumes that the log contains no information of the variables that affect reward generation, while the others assumes that all such variables are present. We consider the middle ground, by assuming that a fraction of these variables are included in the logs. the authors of Tennenholtz et al. (2020) studied a related linear bandit problem where the agent is provided with partial observations collected offline according to a fixed behavioral policy which can be sampled from. These are then used to aid online decision making after the agent observes the full context. In contrast, we consider the case where the agent can only observe the *partial context* at each time and is provided with confounded logs collected from a policy (to which we do not assume sampling access) that is optimal under the full context.

## 2 Bandits with Mean Bounds

We consider the round-based interaction of a learning agent with a stochastic environment through a set of $K_0$ actions (or arms). At each round, the agent chooses one of the provided arms and observes a stochastic reward. To aid its decision making, the agent is also provided with side information in the form of bounds on the mean reward of each arm. The goal of the agent is to choose arms such that the cumulative reward is competitive with respect that obtained by a genie that only chooses the 'best' arm in each round.

In this work, we concentrate on two commonly studied formulations of this problem: Stochastic Multi-armed Bandits and Linear Bandits. We detail the notations, structure of rewards, and notions of the best arm for each of these below.

*Stochastic Multi-armed Bandit:* The agent is provided with $K_0$ arms indexed by the set $[K_0] = \{1, 2, ... K_0\}$, with each arm being associated with a fixed and unknown reward distribution supported over the interval $[0, 1]$. The mean reward of arm $k \in [K_0]$ is denoted by $\mu_k$. For each arm $k \in [K_0]$, the side information is given by a tuple $(l_k(t), u_k(t))$ such that $\mu_k \in [l_k(t), u_k(t)]$ and $l_k(t), u_k(t) \in [0, 1]$. These tuples thus specify upper and lower bounds on the *mean reward* for each of the arms and can be different for each arm. With this knowledge, at round $t$, the agent chooses an arm $A_t$ and observes a reward $Y_t$ sampled from the distribution associated with the chosen arm. We define $k^* = \arg\max_{k \in [K_0]} \mu_k$ be the (unique) best arm with $\mu_{k^*} = \mu^*$ and the genie chooses this arm at each round.

The agent thus aims to minimize its average cumulative regret, which at round $T$ is given by $R_T = \sum_{t=1}^{T} \mu^* - \mathbb{E}[Y_t]$. The expectation here is over the randomness of the rewards and the choice of arms of the agent. If we have that for all $k \in [K_0]$, $l_k = 0, u_k = 1$, then our setting matches that of the standard Multi-armed Bandit with bounded rewards.

*Linear Bandit:* In this case, in each round $t$, the agent is provided a (possibly different) set of action $\mathcal{A}_t \subseteq \mathcal{A}$ sampled from a (possibly infinite) set of actions $\mathcal{A}$ such that $\|\mathcal{A}_t\| = K_0$ . Each arm $a \in \mathcal{A}$ is a vector in $\mathbb{R}^d$. At round $t$, the agent chooses an arm $A_t \in \mathcal{A}_t$ and observes a reward $Y_t = \langle A_t, \theta^* \rangle + \eta_t$ where $\theta^* \in \mathbb{R}^d$ is a fixed unknown vector. The noise $\eta_t$ is a conditionally $\sigma(A_t)$−subgaussian random variable with respect to the filtration $\mathcal{F}_t = \sigma(A_1, Y_1, ..., A_{t-1}, Y_{t-1}, A_t)$. This arm-specific subguassian factor is not revealed to the agent. As in the case above, for all rounds $t$, $Y_t \in [0, 1]$ and the agent is provided with tuples $(l_a, u_a)$ for each $a \in \mathcal{A}$ such that $\theta^T a \in [l_a, u_a]$ and $l_a, u_a \in [0, 1]$. We also assume that $\|\theta^*\| \in [m, M]$ and that $\|a\| = 1$ ($\| \cdot \|$ is the Euclidean norm) for $a \in \mathcal{A}$.

As before, the agent aims to maximize its cumulative reward in order to remain competitive with a genie that chooses the arm with highest mean reward at each round. Equivalently, the agent aims to minimize the regret, which at round $T$ is given by $R_T = \sum_{t=1}^{T} \max_{a \in \mathcal{A}_t} \langle a - A_t, \theta^* \rangle$. In contrast to the setting above, $R_T$ is a random variable due to the randomness in $A_t$. With $l_a = 0, u_a = 1$ for $a \in \mathcal{A}$, our setting becomes that of the standard Linear Bandit with bounded rewards.

For both the settings above, we note three things: *a)* The agent is allowed to use the all historical information and accumulated rewards to inform future arm choices, *b)* The provided bounds *do not* restrict the range of observed rewards, but only their mean, *c)* Our discussion can be easily generalized to bounded rewards supported over any interval $[a, b]$ by appropriate shifting and scaling. Our methods will also apply when bounds were known on the norm of the actions instead of the strict equality in the linear bandit case.

In the standard versions of both the above settings, Optimism-based policies have been well-studied. In the stochastic case, the standard UCB algorithm in Auer et al. (2002) and the KL-UCB algorithm in Cappé et al. (2013) achieve provably optimal performance for specific families of reward distributions. However, with non-trivial mean bounds, it is shown in Zhang & Bareinboim (2017) that one can outperform these methods by leveraging the information that is provided by these bounds. In the linear case, the OFUL algorithm of Abbasi-Yadkori et al. (2011) (or LinUCB of Li et al. (2010)) provides optimal regret guarantees up to logarithmic factors (see Chapter 24 in Lattimore & Szepesvári (2020)). To the best of our knowledge, the linear bandit problem with mean bounds has not been considered before. These naive algorithms that do not use the knowledge of the provided side information will serve as baselines to the methods that we develop.

Before we propose our methods, we will first spend time developing improved concentration bounds for empirical means of random variables given mean bounds. These concentrations will be crucial in analyzing our arm selection policies for both the settings above.

## 3 Improved Concentrations with Mean Bounds

A random variable $X$ is said to be $\sigma-$subgaussian if and only if $\mathbb{E}\left[\exp(s(X - \mathbb{E}[X]))\right] \leq \exp\left(\frac{s^2\sigma^2}{2}\right)$. As a consequence, we have the following Chernoff-Hoeffding concentration inequality for $\sigma-$subgaussian variables:

$$\mathbb{P}(X \geq t) \leq \exp\left(-\frac{t^2}{2\sigma^2}\right).$$

Further, it is well known that any random variable that is bounded in an interval $[a, b]$ is $\frac{b-a}{2}-$subguassian. With no additional information about this variable, we can not improve this factor. Suppose the mean of the random variable were known. We ask the question 'Does this additional information lead to a tighter $(< \frac{1}{2})$ estimate of the subgaussian factor?'. This is important because a tighter subgaussian parameter would lead to faster concentrations of the random variable.

Suppose now that the random variable $X \in [0, 1]$ has mean $m$ that is known. Since it is bounded, the variance of this random variable is bounded by that of a Bernoulli random variable with the same mean. Further, the square of the subgaussian factor for any random variable is an upper bound on its variance (See Lemma 5.4 in Lattimore & Szepesvári (2020)). Therefore, we have that $var(X) \leq m(1-m) \leq sg(\text{Bernoulli}(m))^2 \leq \frac{1}{2}$ where $var(\cdot)$ is the variance and $sg(\cdot)$ is the subgaussian factor of the argument. We will now show that there indeed exist $\sigma \in (m(1-m), \frac{1}{2})$ such that any random variable $X \in [0, 1]$ with mean $m \in (0, 1)$ is $\sigma-$subgaussian (the cases when $m = 0, 1$ are trivial).

For this to hold, we require that

$$\mathbb{E}\left[e^{s(X-m)}\right] \leq m\exp\left(s(1-m)\right) + (1-m)\exp\left(-ms\right) \leq \exp\left(\frac{s^2\sigma^2}{2}\right)$$

$$\implies \sigma^2 \geq \frac{2}{s^2}\log\left((1-m)e^{-ms} + \mu e^{s(1-m)}\right) \geq \max_{s \in \mathbb{R}} \frac{2}{s^2}\log\left(me^{s(1-m)} + (1-m)e^{-ms}\right) \quad (1)$$

Thus, it is sufficient to prove that there exist a unique maximizer to the RHS of Equation 1 and that the achieved maxima is $< \frac{1}{2}$. Below, we give the steps involved in proving this. The full proof is deferred to Appendix A.

We will use a sequence of lemmas to establish our result. For simplicity, we define $f_m(x) = me^{x(1-m)} + (1-m)e^{-mx}$ and begin by proving the following facts:

**Lemma 1.** *For all* $m \in (0, 1)$ *and* $x \in \mathbb{R}$ , $f_m(x) > 1$. *Further,* $f_m(x) = f_{1-m}(-x)$ *and* $\lim_{x\to 0} \frac{2}{x^2}\log\left(f_m(x)\right) = m(1-m)$.

Given these properties of $f_m(x)$, we then prove the following Lemmas:

**Lemma 2.** *The following are true:*

1. *When $m \in (0.5, 1)$, the function $\frac{2}{x^2} \log(f_m(x))$ is not maximized at any $x > 0$.*

2. *For all $m \in (0, 0.5), x > 0$, we have that with $x_1$ as defined in Lemma 15*

   a) *For all $x \in (0, x_1)$, $\frac{2}{x^2} \log(f_m(x)) > m(1 - m)$,*
   b) *For all $x > \frac{2}{m}$, $\frac{2}{x^2} \log(f_m(x)) < m(1 - m)$.*

Combining these two Lemmas, we have our final result:

**Theorem 3.** *Let $X \in [0, 1]$ be a random variable with mean $m \in (0, 1)$, $m \neq 0.5$. Then, $X$ is $\sigma$-subgaussian for $\sigma^2 = \max_{x \in \mathbb{R}} \frac{2}{x^2} \log\left(me^{(1-m)x} + (1-m)e^{-mx}\right)$. Additionally, $\sqrt{m(1-m)} < \sigma < \frac{1}{2}$.*

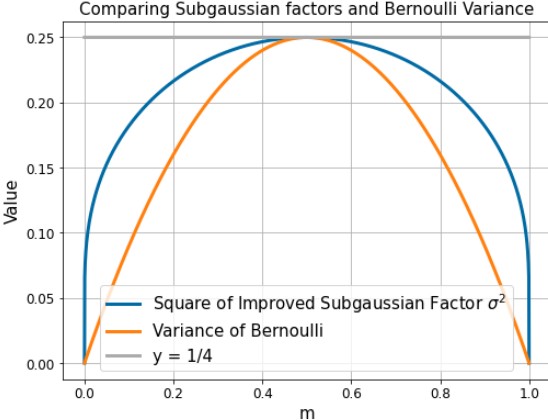

Figure 1: Improved Subgaussian factor vs. Bernoulli variance

The lower bound of $\sqrt{m(1 - m)}$ on $\sigma$ implies that it increases in $(0, 0.5)$ and decreases in $(0.5, 1)$. In Figure 1, we plot the values of $\sigma^2$ and $m(1 - m)$, the variance of a Bernoulli with mean $m$, for different values of $m$ in $(0, 1)$ to verify this trend. This leads to the following corollary:

**Corollary 3.1** (Improved Subgaussian factors with mean bounds)**.** *Let $X$ be a random variable over $[0, 1]$ such that $\mathbb{E}[X] \in [l, u]$ for some $l, u \in [0, 1]$. For any $m \in (0, 1)$, let $c(m) = \max_{s \in \mathbb{R}} \frac{2}{s^2} \log(f_m(s))$ with $f_m(x) = me^{x(1-m)} + (1-m)e^{-mx}$. Then, the following are true:*

1. *If $l > 0.5$, we have that $X$ is $\sqrt{c(l)}$-subgaussian.*

2. *If $u < 0.5$, we have that $X$ is $\sqrt{c(u)}$-subgaussian.*

Thus, bounds on the mean of the random variable that do not contain the worst-case value of 0.5 always lead to improved estimates of its subgaussian factor. We call such mean bounds to be '*non-trivial*'.

## 4 Restricted-set Optimism under Uncertainty for Linear Bandits with Mean Bounds

Now that we know the implications of non-trivial mean bounds on the concentration behavior of a random variable, we move on to studying the effects of such information on the regret of a linear bandit agent. Recall that at each round $t$, the agent is given an action set $\mathcal{A}_t$ of $K_0$ vectors in $\mathbb{R}^d$ sampled from a set $\mathcal{A}$ and must learn to choose actions $A_t \in \mathcal{A}_t$. Observing $Y_t = \langle A_t, \theta^* \rangle + \eta_t$, where $\theta^*$ is an unknown vector and $\eta_t$ a conditionally $\sigma(A_t)$-subgaussian random variable with respect to the filtration generated by all observations up to $t - 1$ and the choice of arm $A_t$, the agent competes with a genie that always picks the arm in the

---

**Algorithm 1** **R**restricted-set **O**ptimism in the **F**ace of **U**ncertainty for **L**inear bandits with mean bounds

---
1: **Inputs:** Action set $\mathcal{A}$, bound tuples $(l_a, u_a)$ for each $a \in \mathcal{A}$
2: *Tightening bounds:* Update the tuples $(l_a, u_a)$ as in Equation 2.
3: **for** t = 1,2,3,... **do**
4:     Receive set $\mathcal{A}_t$, identify $\hat{a}_t$ and $l_{max}(t)$.
5:     *Restricting instantaneous action sets:*
6:     Prune out all arms $a$ with $u_a < l_{max}(t)$.
7:     Form set $\mathcal{A}_r(t)$ defined in Equation 3.
8:     *Reduced exploration:*
9:     Compute $\sigma_t$ as in Equation 4.
10:     With $\mathcal{E}_t$ defined in Equation 5, choose $a_t$ to be $\arg\max_{a \in \mathcal{A}_r(t), \theta \in \mathcal{E}_t} \langle a, \theta \rangle$, observe reward $Y_t$.
11: **end for**

---

provided set with largest mean reward. That is, the agent aims to minimize its regret at round $T$ given by $R_T = \sum_{t=1}^{T} \max_{a \in \mathcal{A}_t} \langle a - A_t, \theta^* \rangle$. As is standard in Linear Bandits, this regret $R_T$ is random due to the randomness in the choice of $A_T$ and thus, we seek to minimize this regret with high probability. Specifically, we use $\delta$ to be the user-specified failure probability and develop a policy that minimizes regret with probability at least $1 - \delta$.

The agent is also provided with tuples $(l_a, u_a)$ for each arm $a \in \mathcal{A}$ which can be used to aid its decision making. Further, it is known that $\|\theta^*\| \in [m, M]$ and that for all $a \in \mathcal{A}, \|a\| = 1$. We propose the Restricted-set Optimism under Uncertainty for Linear Bandits (R-OFUL) algorithm (summarized in Algorithm 1) that uses these mean bounds and the linear structure of rewards to minimize regret. At each round, the agent performs three steps: *a)* Tightening the mean bounds, *b)* Restricting the instantaneous action set, and *c)* Invoking the update similar to the OFUL algorithm of Abbasi-Yadkori et al. (2011) with potentially improved subgaussian factors that leads to reduced exploration. We describe each of these steps below.

**Tightening the Mean Bounds:** Since all rewards are obtained using the same parameter $\theta^*$, bounds on the mean of some action give us non-trivial information about the mean reward of all other actions. This fact is used to tighten the provided upper and lower bounds on arm means. Formally, define $\mathcal{C}_b = \{\theta : \|\theta\| \in [m, M], \forall a \in \mathcal{A}, \langle a, \theta \rangle \in [l_a, u_a]\}$ be the set of feasible parameter vectors. Then, the tight bounds $l_a, u_a$ (we abuse notation by representing both the provided and tighter bounds by the same variables) are computed for each arm $a \in \mathcal{A}$ as

$$l_a \leftarrow \min_{\theta \in \mathcal{C}_b} \langle a, \theta \rangle, \quad u_a \leftarrow \max_{\theta \in \mathcal{C}_b} \langle a, \theta \rangle. \tag{2}$$

After these tight bounds are computed, the set of feasible vectors $\mathcal{C}_b$ is recomputed with these updated versions of the mean bounds.

**Restricting instantaneous action sets:** This phase is carried out after receiving the set $\mathcal{A}_t$ at time $t$ and chooses a subset of arms to be considered at each round. First the agent prunes away deterministically suboptimal arms used $l_{max}(t) = \max_{a \in \mathcal{A}_t} l_a$. This restricts the set of arms in consideration to $\mathcal{A}_p(t) = \{a : u_a \geq l_{max}(t)\}$.

Now we denote by $\hat{a}_t = \arg\max_{a \in \mathcal{A}_t} l_a$ the 'most promising arm' at time $t$ and use $\text{ang}(a, a') = \cos^{-1}\left(\frac{\langle a, a' \rangle}{\|a\|\|a'\|}\right)$ as the angle between the vectors $a$ and $a'$. With $\alpha_t = \cos^{-1}\left(l_{max}(t)/M\right)$, the restricted set of actions at time $t$ is given by

$$\mathcal{A}_r(t) = \{a \in \mathcal{A}_p(t) : \text{ang}(a, \hat{a}_t) \leq 2\alpha_t\} \tag{3}$$

**Reduced Exploration:** In this phase, the agent computes an upper bound on the s.g-factors of arms in the set $\mathcal{A}_r(t)$ and uses this to reduce exploration in its arm selection strategy. Recall that for any $y \in (0, 1)$ we use $c(y) = \max_{s \in \mathbb{R}} \frac{2}{s^2} \log(f_y(s))$ where $f_y(x) = ye^{(1-y)x} + (1-y)e^{-yx}$. For each arm $a \in \mathcal{A}_t$ with (tightened)

mean bounds $l_a, u_a$, we define

$$\psi^2(l_a, u_a) = \begin{cases} c(l_a) & \text{if } l_a > 0.5 \\ c(u_a) & \text{if } u_a < 0.5 \\ 0.25 & \text{otherwise} \end{cases}$$

to be the function that maps the upper and lower bounds to the reduced s.g-factor as a result of Corollary 3.1. Then, upper bound on the s.g-factor of any arm in $\mathcal{A}_r(t)$ is given by

$$\sigma_t = \min \left\{ \psi(m \cos(3\alpha_t), b), \max_{a \in \mathcal{A}_r(t)} \psi(l_a, u_a) \right\}. \tag{4}$$

The agent forms the ridge regression estimate of the parameter $\theta^*$ at time $t$ as $\hat{\theta}_t = \overline{V}_t^{-1} \left( \sum_{n=1}^t a_n Y_n \right)$ with $\overline{V}_t = \sum_{n=1}^t a_n a_n^T + \lambda I_d$. Here $I_d$ is the $d$-dimensional identity matrix and $\lambda > 0$ is a regularization parameter. It then defines the following set around $\theta^*$:

$$\mathcal{C}_t = \left\{ \theta \in \mathcal{C}_b : \left\| \hat{\theta}_t - \theta \right\|_{\overline{V}_{t-1}} \leq \beta_{t-1}(\delta) \right\}, \quad \sqrt{\beta_t(\delta)} = \sqrt{\lambda} M + \gamma \sqrt{2 \log \left( \frac{1}{\delta} \right) + d \log \left( 1 + \frac{t}{d\lambda} \right)}. \tag{5}$$

Here $\gamma = \max_{n \leq t} \sigma_n$ and $\delta$ is the user-specified failure probability. The choice of arm $A_t$ is given by

$$A_t = \arg\max_{a \in \mathcal{A}_r(t), \theta \in \mathcal{C}_t} \langle a, \theta \rangle. \tag{6}$$

This selection rule is similar to that of vanilla OFUL in Abbasi-Yadkori et al. (2011). The difference here, we restrict *a)* the set $\mathcal{C}_t$ with the feasible set $\mathcal{C}_b$, *b)* the set of arms that can be played at time $t$ with $\mathcal{A}_r(t)$ and *c)* the confidence width $\beta_t(\delta)$ using the tighter subgaussian estimate of $\gamma$.

### 4.1 Key Ideas

**1. Deriving the restricted set:** The boundedness and linearity of rewards implies that if there exists an arm that promises a high mean (in our case, $\hat{a}_t$), its angle with the parameter $\theta^*$ is upper bounded. This is captured by the quantity $\alpha_t$. Further, it also implies that the angle between the best arm in the set $\mathcal{A}_r(t)$ and $\theta^*$ can be no more than $\alpha_t$. Combining these, we get that the best arm lies in a cone of angle $2\alpha_t$ around $\hat{a}_t$.
**2. Tight s.g-factors:** Since the goal is to minimize regret, the worst-case arm that can be played (one with the lowest mean reward) in $\mathcal{A}_r(t)$ is one that is at an angle $3\alpha_t$ away from $\theta^*$. As rewards are bounded, this lower bound on the mean reward of the worst arm can be translated into an upper bound on the s.g-factor of arms in $\mathcal{A}_r(t)$ using the function $\psi$. If the side information of arms in $\mathcal{A}_r(t)$ can provide sharper estimates on this upper bound, we use those instead. This is reflected in our definition of $\sigma_t$ in Equation 4

### 4.2 Regret of R-OFUL

We now present our high-probability regret guarantees for Algorithm 1.

**Theorem 4.** *With probability at least $1 - \delta$, the regret of R-OFUL suffers a worst-case regret of*

$$R_t \leq \sqrt{8 d t \beta_{t-1}(\delta) \log \left( 1 + \frac{t}{d\lambda} \right)}.$$

*Proof Sketch.* First, we establish the following lemma:

**Lemma 5.** *Let $a_t^* = \arg\max_{a \in \mathcal{A}_t} \langle a, \theta^* \rangle$ be the best arm at time $t$. Then, $a_t^* \in \mathcal{A}_r(t)$. Further, for any arm $a \in \mathcal{A}_t$, the s.g-factor $\sigma(a)$ satisfies $\sigma_a \leq \sigma_t$.*

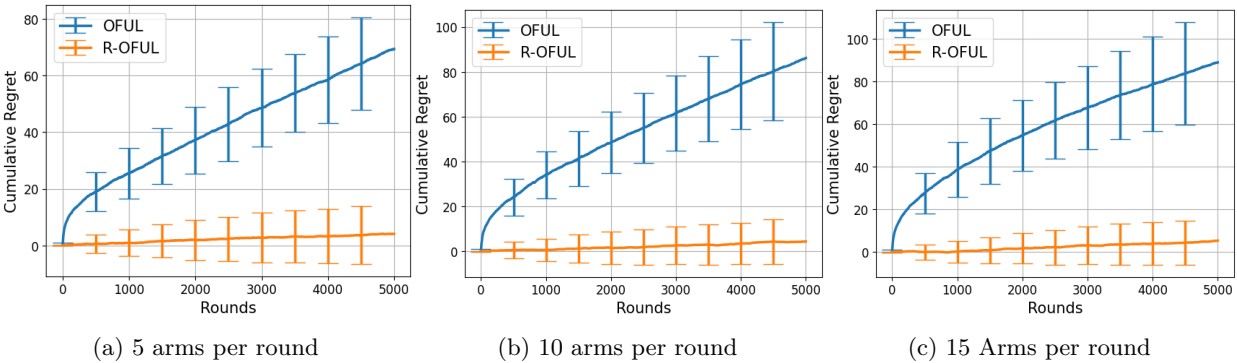

(a) 5 arms per round  (b) 10 arms per round  (c) 15 Arms per round

Figure 2: Comparing R-OFUL (Algorithm 1) with vanilla OFUL with arms in $\mathbb{R}^{10}$ and bounded rewards. Results are averaged over 200 runs and error bars for one standard deviation are displayed. R-OFUL restricts arms and only chooses between 2-3 arms per round on average and thus, its average regret is comparable over all three figures, while OFUL suffers regret that grows with the number of arms per round. We also observe that R-OFUL computes arm updates $6 - 6.7\times$ faster on average.

Using this lemma, we generalize the concentration arguments of Abbasi-Yadkori et al. (2011) to prove that with probability at least $1 - \delta$, the parameter vector satisfies $\theta^* \in \mathcal{C}_b$. The result then follows using the standard arguments for bounding regret in linear bandits. □

The full proof can be found in Appendix B.

### 4.3 Comparisons and Discussions

**R-OFUL vs OFUL:** The regret of the OFUL algorithm is of the order $\tilde{\mathcal{O}}\left(\sqrt{dt\sigma_{max}(d + \log(1/\delta))}\right)$ with $\sigma_{max} = {}^{(b-a)}/_2$ as the universal upper bound on the s.g-factor of the arms in the set $\mathcal{A}$. If no informative bounds are present (the worst case) our R-OFUL algorithm matches the performance of OFUL exactly. However, richer side information can lead to a constant factor improvement in the finite-time regret guarantees as $\gamma \leq \sigma_{max}$. Further, in cases where $\|\mathcal{A}_r(t)\| = 1$, the instance-dependent regret of R-OFUL is 0, while any linear bandit algorithm that does not perform this arm restriction will suffer non-zero regret on average.

We also improve on the computation complexity of OFUL. Specifically, to compute $A_t$ (Equation 6), the usual practice is to solve the optimization problem for all $a$ individually for all arms and then choose one with the maximum objective value. If the mean bounds are such that $|\mathcal{A}_r(t)| < |\mathcal{A}_t|$, that is, if the set of arms is restricted by R-OFUL, then, so is the number of optimization sub-problems to solve, which speeds up computation.

**Optimality:** We study a generalization of OFUL which is known to be optimal up to logarithmic factors in the worst case. However, it is known that optimality in general linear bandit problems is achieved by complex non-optimistic policies. The study of regret lower bounds and policies that match them are left open.

**Empirical Evaluations:** We compare the regret performance of R-OFUL with the vanilla OFUL algorithm empirically in Figure 2. For this, we sample a random set of 100 arms in $\mathbb{R}^{10}$ and sample $5, 10, 15$ of these in each round. We set $\theta^*$ as the vector $\left[0, \frac{1}{10}, \frac{2}{10}, \ldots \frac{9}{10}\right]$ and normalize it. To generate rewards, for each arm $a \in \mathcal{A}_t$, we first sample a Gaussian random variable with mean $\langle a, \theta^* \rangle$ and variance 0.8, and clip this to be in $[0, 1]$ ($[-1, 0]$) if $\langle a, \theta^* \rangle > 0 (\leq 0)$ to form our bounded rewards. The upper and lower bounds on the rewards are generated separately to be away from the mean by a uniform random variable in $[0, 0.5]$ and these are also clipped the same way as the rewards. That is, arms with positive (non-positive) mean reward are forced to have bounds in $[0, 1]$ ($[-1, 0]$). The subgaussian upper bound for OFUL is provided as 0.5, the worst-case subgaussian factor for any random variable bounded in either $[0, 1]$ or $[-1, 0]$ while R-OFUL computes tighter subgaussian factors based on the mean bounds and uses $\gamma$ as in Equation 5 to choose arms. We also use the norm bounds on $\theta^*$ to $m = 0, M = 1$. We average our results over 200 independent runs. We see that R-OFUL consistently achieves much lower regret than the vanilla variant. Further, we also observe that

---

**Algorithm 2 GL**obal **U**nder-**E**xplore (GLUE) for MABs with Mean Bounds

---

1: **Inputs:** Upper and lower bounds for each arm $k \in [K_0] : u_k, l_k$.
2: **Pruning Phase:**
3: Define $l_{max} = \max_{k \in K_0} l_k$ and eliminate all arms $i \in [K_0]$ with $u_i < l_{max}$.
4: Reindex the remaining arms to be in $\{1, 2, ..., K\}$.
5: **Learning Phase:**
6: Set UCB index scaling parameters $\psi_k$ for each arm as in Equation (7).
7: **for** $t = 1, 2, 3, ...$ **do**
8:     Play arm $A_t = \arg\max_{k \in [K]} U_k(t-1)$ and observe reward $Y_t$.
9:     Increment $T_{A_t}(t)$ and update $\hat{\mu}_{A_t}(t)$.
10:     Update $U_k(t)$ as in Equation (8).
11: **end for**

---

empirically, R-OFUL restricts arms and only chooses between 2-3 arms per round on average. This leads to a $6 - 6.7\times$ computation speed up compared to vanilla OFUL.

## 5 Global Under-exploration for Stochastic Multi-armed Bandits with Mean Bounds

Now we move on to the stochastic Multi-armed Bandit (MAB) setting. We recall that in this case, there is no linear structure on the rewards. At each time, the agent picks one of $K_0$ arms, and observes a reward that is randomly sampled from the distribution that is associated with this arm. The agent can also access the tuples $(l_k, u_k)$ for all arms $k \in [K_0]$ to inform its choices. At round $T$, the agent aims to minimize the expected cumulative regret $R_T = \sum_{t=1}^{T} \mathbb{E}[\mu^* - Y_t]$.

Prior work in the Zhang & Bareinboim (2017); Combes et al. (2017) have investigated this problem before, the for Bernoulli rewards, and the latter as a structured bandit problem. The B-KL-UCB algorithm of the former sets arms indices as the clipped version (using the provided bounds) of the the standard KL-UCB indices from Cappé et al. (2013). The latter proposes OSSB for general reward distributions where the agent decides arms at each round by solving a distribution-dependent, semi-infinite optimization problem. The authors also prove that OSSB achieves asymptotically optimal regret for Bernoulli and Gaussian rewards when reward distributions are known apriori. In the special case of Bernoulli rewards, OSSSB and B-KL-UCB are equivalent (see Appendix D), and are thus optimal.

However, it is often not practical to assume that the reward distributions are known apriori. Further, even with this knowledge, it is unclear how computationally tractable the per-round optimization problem of OSSB is. While the Bernoulli version in B-KL-UCB is still applicable in this case, the KL-UCB indices for each arm in each round are themselves optimization problems with no known closed form solution. As the number of arms grow, computing these indices efficiently poses a challenge.

The UCB algorithm of Auer et al. (2002) provides an index-based solution that is inexpensive to compute, albeit at the cost of the optimal regret performance. Specifically, it is well known that in the vanilla MAB setting, KL-UCB always outperforms UCB for bounded reward distributions, with larger gains when the mean rewards are closer to the extremities. This is mainly due to the fact that UCB chooses the worst-case exploration rate for any bounded distribution, while KL-UCB adapts its rate according to the (unknown) value of the mean rewards. Works such as Liu et al. (2018) address the computation issue by using a set of semi-distance functions to boost the UCB index. This leads to a trade-off between optimal performance and efficient compute.

We propose Global Under-Exploration (GLUE) for Stochastic MABs with Mean Bounds in Algorithm 2. It draws inspiration from KL-UCB, adapting its exploration rate to the location of the arm means using the provided mean bounds but like UCB, provides arm indices that are inexpensive to compute. We note that unlike B-KL-UCB, GLUE is *not* the clipped version of vanilla UCB. In particular, GLUE is *not optimistic*, i.e, its arm indices are not high-probability upper bounds to the true mean of the arm. The algorithm works in two phases:

**Pruning Phase:** Before the start of the online learning process, GLUE examines the provided mean bounds of each of the $K_0$ arms are prunes out any deterministically suboptimal arms. Specifically, we compute $l_{max} = \max_{k \in [K_0]} l_k$ and discard any arm $k$ with $u_k < l_{max}$. Such a strategy is also followed by B-KL-UCB of Zhang & Bareinboim (2017).

**Learning Phase:** Let $[K] = \{1, 2, ..., K\}$ be the (re-indexed) set of pruned arms, with $K \leq K_0$. For each of the arm $k \in [K]$, we compute

$$\sigma_k^2 = \begin{cases} c(l_k) & \text{if } l > 0.5 \\ c(u_k) & \text{if } u < 0.5 \\ 0.25 & \text{otherwise} \end{cases}, \quad \psi_k = \begin{cases} \sigma_{k_{max}} & \text{if } l_{k_{max}} > 0.5 \\ \sigma_k & \text{otherwise.} \end{cases} \tag{7}$$

Here, $k_{max} = \arg\max_{k \in [K]} l_k$, and $c(y) = \max_{s \in \mathbb{R}} \frac{2}{s^2} f_y(s)$ with $f_y(x) = ye^{(1-y)x} + (1-y)e^{-yx}$. Let $T_k(n) = \sum_{t=1}^n \mathbb{1}\{A_t = k\}$ and $\hat{\mu}_k(n) = \frac{1}{T_k(n)} \sum_{t=1}^n Y_t \mathbb{1}\{A_t = k\}$ be the number of plays and the empirical mean reward obtained from arm $k$ up to time $n$ respectively. The index of arm $k$ at time $t$ is then set to be

$$U_k(t) = \min \left\{ u_k, \hat{\mu}_k(t) + \sqrt{\frac{2\psi_k^2 \log(f(t+1))}{T_k(t)}} \right\} \text{ with } f(t) = 1 + t\log^2(t). \tag{8}$$

We observe two key things here. First, for each arm $k$, $U_k(t)$ is set using the quantity $\psi_k$ rather than the true subgaussian factor $\sigma_k^2$. When $l_{max} > 0.5$, $\psi_k$ for *all arms* is set as the subgaussian factor of the arm with the largest lower bound $k_{max}$ which is strictly lower than $\sigma_k^2$ for $k \neq k_{max}$. Thus in general, $\psi_k \leq \sigma_k^2$. Second, arm indices are clipped at the respective upper bounds, which is sensible as these indices are a representation of our belief of true arm means. We also note that using $\psi_k = 0.25$ and setting $U_k(t)$ without clipping at $u_k$ recovers the standard Upper Confidence Bound (UCB) algorithm in Auer et al. (2002).

## 5.1 Key Ideas

**1. Sharper subgaussian factors:** As we saw in Section 3, bounds on the mean of a random variable (here, the arm rewards), provide tighter estimates of its subgausssian factor (Corollary 3.1). This is reflected in our definition of $\sigma_k$ in Equation 7.

**2. Underexploring suboptimal arms using $\psi_k$:** Consider a standard MAB instance with $K$ arms where it is known that arm $k$ provides rewards with a subgaussian factor of $\sigma_k^2$. The standard UCB algorithm would then explore arm $k$ at the rate dictated by the corresponding subgaussian factor. Now suppose additionally, that it was known that the (unknown) *best arm* produced rewards with subgaussian factor $\sigma_{best}^2$. We construct the algorithm $\mathcal{A}$ which sets the index of arm $k$ analogous to UCB, but with exploration rate dictated by $\min\{\sigma_k^2, \sigma_{best}^2\}$. As a result, $\mathcal{A}$ potentially assumes *incorrect* s.g-factors for suboptimal arms. Consequently, the indices of $\mathcal{A}$ are no longer upper confidence bounds to the arm means, i.e., $\mathcal{A}$ is *not optimistic*. The key difference in the dynamics of UCB and $\mathcal{A}$ is that the latter potentially explores suboptimal arms *less frequently* than the former. This would lead to $\mathcal{A}$ incurring lower regret than standard UCB.

In GLUE, $\min\{\sigma_k^2, \sigma_{best}^2\}$ is analogous to $\psi_k$. Specifically, when $l_{max} > 0.5$, the side information allows us to estimate an upper bound on the subgaussian factor of the *best arm*. We note that this upper bound is computed using the arm $k_{max}$; the identity of the best arm is still unknown. Thus, the regret minimization problem remains non-trivial in this case.

## 5.2 Regret Upper Bounds for GLUE

In this section, we study the regret performance of GLUE. We regard improvements from pruning as trivial and analyze regret for the set of arms that remain after pruning. We assume without loss of generality that *after pruning* the arms are indexed in non-decreasing order of mean, i.e., $\mu^* = \mu_1 > \mu_2 \geq ... \geq \mu_K$, where $K \leq K_0$. We also define the sub-optimality gap of an arm $k$ as $\Delta_k = \mu^* - \mu_k$. The following theorem bounds the regret of GLUE, and also presents its asymptotic regret scaling which is an upper bound to $\limsup_{n \to \infty} \frac{R_n}{\log(n)}$. This quantity captures the long-term (logarithmic) contribution of arms towards regret.

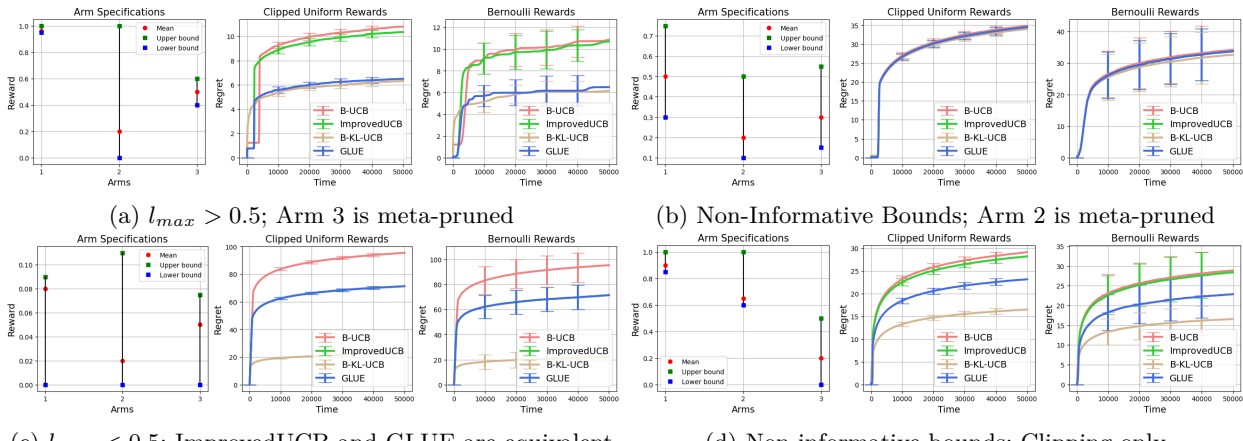

(a) $l_{max} > 0.5$; Arm 3 is meta-pruned

(b) Non-Informative Bounds; Arm 2 is meta-pruned

(c) $l_{max} < 0.5$; ImprovedUCB and GLUE are equivalent

(d) Non-informative bounds; Clipping only

Figure 3: Empirical validation for Stochastic MABs with Mean Bounds under Clipped Uniform and Bernoulli rewards: Each row corresponds to a different specification of arm means and mean bounds shown in the left subplot. Regret performance under clipped uniform and Bernoulli rewards are shown in the latter two. The regret is averaged over 200 runs and error bars of one standard deviation are shown. In Figure 3a, for each arm, $\psi_k = \sigma_1$. In Figure 3b, the bounds reveal no non-trivial information, however, Arm 2 is meta-pruned. In Figure 3c, we set $\psi_k = \sigma_k$ since $l_{max} < 0.5$ and thus, ImprovedUCB and GLUE coincide. In Figure 3d, the bounds do not provide non-trivial information about subgaussian factors and are only used to clip rewards. B-UCB, UCBImproved and GLUE compute arm choices $11\times$ faster than B-KL-UCB on average.

**Theorem 6.** *Let* $K_1(\delta) = \{k \in [K] : \mu^* > u_k + \delta, k > 1\}$ *and* $K_2(\delta) = \{k \in [K] : \mu^* \leq u_k + \delta, k > 1\}$, *for any* $\delta > 0$. *Then, the expected cumulative regret of Algorithm 2 satisfies* $R_n \leq \inf_{\delta>0} R_n(\delta)$ *with*

$$R_n(\delta) \leq \sum_{k \in K_1(\delta)} \frac{5\psi_1^2 \Delta_k}{(\max\{\delta, \mu^* - u_k\})^2} + \sum_{k \in K_2(\delta)} \left( \Delta_k(1 + q(n)) + \frac{5\psi_1^2 + 2\psi_k^2 \log f(n)}{\Delta_k} \right).$$

*Here, the function* $q(n) \sim \Theta\left(\log(f(n))^{\frac{2}{3}}\right)$. *Further, the asymptotic regret of GLUE satisfies*

$$\limsup_{n \to \infty} \frac{R_n}{\log(n)} \leq \sum_{k \in K_2(0)} \frac{2\psi_k^2}{\Delta_k}. \tag{9}$$

*Proof Sketch for Theorem 6.* As is usual in the regret analysis of stochastic MABs, we upper bound the number of times a suboptimal arm is played, which translates to a bound on the cumulative regret. In contrast however, we can not rely on the UCB property of the indices any longer.

Recall that we only need to consider the set of $K$ arms that remain after pruning. Theorem 3 and Corollary 3.1 gives us that $\sigma_k$ is an upper bound on the true s.g-factor for arm $k$. We first show that the best arm $k^*$ is always $\psi_k-$subgaussian. Then, we establish that the asymptotic contribution to regret of any arm in $K_1(\delta)$ is a constant. We deem these arms to be "`meta-pruned`". For $K_2(\delta)$, we show that the use of pseudo-variances is sufficient to explore these arms at a logarithmic rate. The theorem then follows by combining these two results, with the full proof in Appendix C. □

### 5.3 Comparison and Discussions

**Regret Upper Bounds: GLUE vs Existing Algorithms:** We compare the regret upper bound of GLUE to a slew of baselines in Table 1. Here, B-UCB is the clipped version of the vanilla UCB algorithm of Auer et al. (2002). It is equivalent to using GLUE with $\psi_k = 0.25$, the worst-case subgaussian factor for each arm $k$. Further, $d(p, q) = p \log(p/q) + (1 - p) \log((1-p)/(1-q))$ is the Bernoulli Kullback-Liebler divergence.

Table 1: Asymptotic Regret Comparisons: Columns list the algorithm, the (bounded) distribution of rewards, whether it displays meta-pruning, and the **asymptotic** regret upper bound.

| Algorithm | Reward dist. (bounded) | Meta-pruning | Asymptotic Regret |
|---|---|---|---|
| UCB | Any | No | $\sum\limits_{k\in[K],k\neq 1} \frac{2}{\Delta_k} \cdot \frac{(b-a)^2}{4}$ |
| B-UCB | Any | Yes | $\sum\limits_{k\in K_2(0)} \frac{2}{\Delta_k} \cdot \frac{(b-a)^2}{4}$ |
| KL-UCB | Any | No | $\sum\limits_{k\in[K],k\neq 1} \frac{\Delta_k}{d(\mu_k,\mu^*)}$ |
| B-KL-UCB | Any | Yes | $\sum\limits_{k\in K_2(0)} \frac{\Delta_k}{d(\mu_k,\mu^*)}$ |
| OSSB | Bernoulli | Yes | $\sum\limits_{k\in K_2(0)} \frac{\Delta_k}{d(\mu_k,\mu^*)}$ |
| GLUE | Any | Yes | $\sum\limits_{k\in K_2(0)} \frac{2\psi_k^2}{\Delta_k}$ |

The vanilla UCB and KL-UCB algorithms do not display meta-pruning as they do not use the mean bounds. The regret bound of OSSB in Combes et al. (2017) matches that of B-KL-UCB for Bernoulli rewards (see Appendix D) and is optimal.

The UCB, B-UCB and GLUE algorithms require $\mathcal{O}(K_0)$ compute per iteration in the worst-case (when no arms are pruned). KL-UCB, B-KL-UCB (and OSSB for Bernoulli rewards) require $\mathcal{O}(\alpha_{opt}K_0)$ computational complexity, with $\alpha_{opt}$ the cost of solving the index optimization problem for each arm.

**Uncertain mean bounds:** Our focus in this work has been the case when the provided mean bounds hold with probability 1. The immediate follow-up would be the case where these bounds only hold for each arm $k$ with some probability $p_k < 1$. Practically, such settings occur when we are given historical data about plays from each arm (this is the setting studied in Shivaswamy & Joachims (2012)). The following corollary gives a horizon-dependent regret bound for GLUE in this case.

**Corollary 6.1.** *Let $\mu_k \in [l_k, u_k]$ hold with probability $(1-p_k)$ for each arm $k \in [K]$. Then by time $n$, GLUE suffers an additional regret of $n \sum\limits_{k\in[K]} p_k$ over Theorem 6.*

If we know the horizon to be $T$ and each arm is observed $\mathcal{O}(\log T)$ times in the provided history, using standard Chernoff-type arguments, we easily see that $\sum_{k\in[K]} p_k = O(1/T)$. Thus, the additional regret incurred by time $T$ is simply a constant. However, the cases when the number of samples for an arm are *sub-logarithmic* in the horizon and that with an *unknown horizon* are left open.

**Lack of global under-exploration in R-OFUL:** In the stochastic MAB with bounds, we used GLUE to explore *all* arms at rate $\psi_k^2$ — the subgaussian factor of the best arm (when $l_{max} > 0.5$). This was possible because the estimate of the best arm remained a $\psi_k^2$-subgaussian random variable, and was thus explored at the correct rate. In the linear case, the estimates for each arm uses all samples collected so far in $\hat{\theta}_t$, the ridge regression estimate of $\theta^*$. The estimate of the best arm is thus no longer $\psi_k^2$-subgaussian and under-exploring arms in the set $\mathcal{A}_r(t)$ using $\psi(l_{max}, b)$, for example, would lead to poor estimates even for the best arm and thus larger regret.

**Empirical Comparisons:** We compare GLUE with B-UCB, ImprovedUCB (GLUE with $\sigma_k$ instead of $\psi_k$ in $U_k(t)$) and B-KL-UCB for clipped uniform distributions (drawn from a uniform distribution in an interval around the mean that is fully contained in $[0,1]$) and Bernoulli rewards. We chose not to compare with OSSB since it is unclear how the associated optimization problem can be solved for this distributions. We drop UCB and KL-UCB since the clipped variants in B-UCB and B-KL-UCB, respectively, are never worse then the vanilla counterprats. All plots are averaged over 200 independent runs.

When $l_{max}$ is close to 1, we see that GLUE outperforms the optimistic UCB-based baselines (B-UCB and ImprovedUCB) and is comparable to $B-KL-UCB$. We note that this is the case where KL variants enjoy maximum improvements over UCB variants, but GLUE competes with the optimal in this case. In the case when all upper bounds are lesser than 0.5, GLUE matches the performance of ImprovedUCB, thus improving over B-UCB. However, since there is no information about the best arm in this case, B-KL-UCB outperforms

GLUE. We also note that each of B-UCB, ImprovedUCB and GLUE compute each iteration $11\times$ faster than B-KL-UCB on average.

# 6 A Use Case: Learning from Confounded Logs

Up to this point, we have assumed that the learning agent has access to a tuple of mean bounds for each arm before the start of the online learning process. In this section, we describe a practically motivated scenario in which such mean bounds arise naturally. Specifically, we consider the task of extracting non-trivial bounds on means from partially confounded data from an optimal oracle. We show that under mild assumptions, this problem leads to the stochastic MAB problem (and a Linear Bandit) with mean bounds.

## 6.1 Confounded Logs for Stochastic Multi-armed Bandits

**The Oracle Environment:** We consider a contextual environment, where nature samples a context vector $(z, u) \in \mathcal{C} = \mathcal{Z} \times \mathcal{U}$ from an unknown but fixed distribution $\mathbb{P}$. We assume that sets $\mathcal{Z}$ and $\mathcal{U}$ are both *discrete*. At each time any of the $K_0$ actions from the set $\mathcal{A} = \{1, 2, ..., K_0\}$ can be taken. The reward of each arm $k \in \mathcal{A}$ for a context $(z, u) \in \mathcal{C}$ has mean $\mu_{k,z,u}$ and support $[0, 1]$ (with appropriate shifting and scaling, this can be generalized for any finite interval $[a, b]$). For each context $(z, u) \in \mathcal{C}$, let there be a *unique best arm* $k_{z,u}^*$ and let $\mu_{z,u}^*$ be the mean of this arm.

**Confounded Logs:** The oracle observes the complete context $(z_t, u_t) \in \mathcal{C}$ at each time $t$ and also knows the optimal arm $k_{z_t,u_t}^*$ for this context. She picks this arm and observes an independently sampled reward $y_t$ with mean value $\mu_{k_{z_t,u_t}^*, z_t, u_t} = \mu_{z_t,u_t}^*$. She logs the information in a data set while omitting the partial context $u_t$. In particular, she creates the data set $\mathcal{D} = \{(z_t, k_t, y_t) : t \in \mathbb{N}\}$.

**The Agent Environment:** A new agent is provided with the oracle's log. In this paper, we consider the *infinite data* setting. The agent makes sequential decisions about the choice of arms having observed the context $z_t \in \mathcal{Z}$ at each time $t = \{1, 2, ...\}$, while the part of the context $u \in \mathcal{U}$ is *hidden* from this agent. Let $a_t$ be the arm that is chosen, $z_t$ be the context, and $Y_t$ be the reward at time $t$. Define the average reward of arm $k \in \mathcal{A}$ under the *observed* context $z \in \mathcal{Z}$ as $\mu_{k,z} = \sum_{u \in \mathcal{U}} \mu_{k,z,u} \mathbb{P}(u|z)$.

The optimal reward of the agent under context $z \in \mathcal{Z}$ is defined as $\mu_z^* = \max_{k \in \mathcal{A}} \mu_{k,z}$. The agent aims to minimize its cumulative regret for each context separately. The cumulative reward for each $z \in \mathcal{Z}$ at time $T$ is defined as: $R_z(T) = \mathbb{E}\left[\sum_t \mathbb{I}(z_t = z)(\mu_z^* - Y_t)\right]$

### 6.1.1 Transferring Knowledge through Bounds

The agent is interested in the quantities $\mu_{k,z}$, the mean reward of arm $k$ under the partial context $z$, for all arms $k \in \mathcal{A}$ and contexts $z \in \mathcal{Z}$, to minimize its cumulative regret. As the oracle only plays an optimal arm after seeing the hidden context $u$, the log provided by the oracle is biased and thus $\mu_{k,z}$ can not be recovered from the log in general. However, it is possible to extract non-trivial upper and lower bounds on the average $\mu_{k,z}$. In a binary reward and action setting, similar observations have been made in Zhang & Bareinboim (2017). Alternative approaches to this problem, that include assuming bounds on the inverse probability weighting among others, are discussed in Section 1.1.

Our assumption below is different, and in a setting where there are *more than two arms*. We specify that the logs have been collected using a policy that plays an *optimal arm* for each $(z, u) \in \mathcal{C}$, but do not explicitly impose conditions on the distributions. Instead, in Assumption 1, we impose a separation condition on the *means* of the arms conditioned on the full context. We define:

**Definition 1.** *Let us define $\underline{\delta}_z, \overline{\delta}_z$ for each $z \in \mathcal{Z}$ as follows:*

$$\underline{\delta}_z \leq \min_{u \in \mathcal{U}} \left[ \mu_{z,u}^* - \max_{k \in \mathcal{A}, k \neq k_{z,u}^*} \mu_{k,z,u} \right], \quad \overline{\delta}_z \geq \max_{u \in \mathcal{U}} \left[ \mu_{z,u}^* - \min_{k \in \mathcal{A}, k \neq k_{z,u}^*} \mu_{k,z,u} \right].$$

Thus, for each observed context $z$, these quantities specify the sub-optimality gaps between the best and second best arms, as well as the best and the worst arms, respectively and which hold uniformly over the hidden contexts $u \in \mathcal{U}$.

**Assumption 1** (Separation Assumption). *The vectors $\{\underline{\delta}_z, \overline{\delta}_z : z \in \mathcal{Z}\}$ are provided as a part of the log. Additionally, for each $z \in \mathcal{Z}$, we have that $\underline{\delta}_z > 0$.*

**Remarks on Model and Assumption 1:**

• *Relation to Gap in Agent Space:* We note that these gaps in the *latent space* do not allow us to infer the gap in *the agent space.* However, due to optimal play by oracle, these gaps help in obtaining mean bounds for *all* arms, even those which have not been recorded in the log (Theorem 7).

• *Interpretation:* The gap $\underline{\delta}_z$ gives us information about the hardest arm to differentiate in the latent space when the full context was observed ($\overline{\delta}_z$ analogously is for the easiest arm to discredit). We also note that our approach can still be applied when the trivial bound $\overline{\delta}_z = 1$, or universal bounds not depending on context $\overline{\delta} \geq \overline{\delta}_z$, $\underline{\delta} \leq \underline{\delta}_z$ are used, leading to bounds that are not as tight.

• *Motivating Healthcare Example:* Often, hospital medical records contain information $z_t$ about the patient, the treatment given $k_t$, and the corresponding reward $y_t$ that is obtained (in terms of patient's health outcome). However, a *good doctor* looks at some other information $u_t$ (that is not recorded) during consultation and prescribes the best action $k_t$ under the full context $(z_t, u_t)$. If one is now tasked with developing a machine learning algorithm (an agent) to automate prescriptions given the medical record $z$, this agent algorithm needs to find the best treatment $k(z)$ on average over $\mathbb{P}(u|z)$. Furthermore, the gaps on treatments effects can potentially be inferred from other data sets like placebo-controlled trials.

• *Existing alternate assumptions:* Studies in Yadlowsky et al. (2022); Zhao et al. (2019) impose conditions on bounded sensitivity of the effects with respect to the hidden/unrecorded context (effectively, that this context does not significantly alter the effect). In our work, we explore the other alternative where the treatment has been chosen optimally with respect to the hidden/unrecorded context, and allows strong dependence on this hidden context.

**Quantities computed from log data:** The following quantities can now be computed by the agent from the observed log for each arm $k \in [K_0]$ and each visible context $z \in \mathcal{Z}$:

**1.** $p_z(k)$ : The probability of picking arm $k$ under each context $z$ is denoted as $p_z(k)$. Mathematically, $p_z(k) = \sum\limits_{u \in \mathcal{U}: k = k_{z,u}^*} \mathbb{P}(u|z)$.

**2.** $\mu_z$ : The average reward observed under each context $z$ is defined as $\mu_z$. It can be computed by averaging observed rewards for all the entries with context $z$. This can be written as $\mu_z = \sum\limits_{u \in \mathcal{U}} \mu_{z,u}^* \mathbb{P}(u|z)$.

**3.** $\mu_z(k)$: The contribution from arm $k$ to the average reward $\mu_z$. Thus we have that $\mu_z(k) = \sum\limits_{u \in \mathcal{U}: k = k_{z,u}^*} \mu_{z,u}^* \mathbb{P}(u|z)$.

**4.** $K_>(k,z)$: Finally, we identify the set of arms with "large rewards" $K_>(k,z) = \{k' \in [K_0] : k' \neq k, \mu_z(k') > \overline{\delta}_z p_z(k')\}$ for each $k \in [K]$ and $z \in \mathcal{Z}$.

To see that these quantities can indeed be inferred, assume that the logs are given as an infinite table with columns $Z, A, Y$. Now, we collapse rows that share $Z = z, A = k$ into a single row with $Y$ now being the mean reward of all such rows in the original table.

With this new finite table, to compute $p_z(k)$, we fix all rows with $Z = z$ and compute the fraction of these that have $A = k$. $\mu_z(k)$ is the average reward in all rows that have $Z = z, A = k$, and $\mu_z = \sum_{k \in \mathcal{A}} \mu_z(k)$. Finally, the set $K_>(k,z)$ can be computed from $\mu_z(k), p_z(k)$ and the upper bound on the latent suboptimality gap $\overline{\delta}_z$.

**Bounds in terms of computed quantities:** Using the quantities defined above, the following theorem describes how one can compute lower and upper bounds on the arm rewards in the agent space.

**Theorem 7.** *The following statements are true for each $k \in [K_0]$ and $z \in \mathcal{Z}$:*

*1.* ***Upper Bound:*** *$\mu_{k,z} \leq u_{k,z} := \mu_z - \underline{\delta}_z(1 - p_z(k))$.*

2. **Lower Bound:** $\mu_{k,z} \geq l_{k,z} := \mu_z(k) + \sum_{k' \in K_>(k,z)} \left[ \mu_z(k') - \bar{\delta}_z p_z(k') \right].$

Note that these bounds can be provided for all arms $k \in [K_0]$ and contexts $z \in \mathcal{Z}$. Specifically, bounds can also be extracted for the arms that are never played in the log. This comes as a result of the gaps defined in Definition 1. Proving this results requires a careful use of the total probability theorem which can be found in Appendix E

**Remarks:**
*1. Finite Log:* While we study the case of infinite logs, our approach can be readily extended to finite logs by replacing the quantities $\mu_z, \mu_z(k), p_z(k)$ with empirical estimates and using standard Chernoff bounds to augment the bounds in Theorem 7 to hold with a probability which is a function of the size of the log. Then, the performance of the learning algorithm is as in Corollary 6.1 and the discussion that follows.
*2. Distribution shifts:* Our results can also be used in the case where under a fixed visible context, the conditional distribution of the hidden contexts changes from oracle to the learner. In particular, for some $z \in \mathcal{Z}$, suppose that $\psi_z \geq \max_{u \in \mathcal{U}} |\mathbb{P}_o(u|z) - \mathbb{P}_l(u|z)|$, where $\mathbb{P}_o$ and $\mathbb{P}_l$ are used to differentiate the oracle and learner environments respectively. Using $\psi_z$, we can readily modify our results in Theorem 7 to produce upper and lower bounds for arm rewards under the context $z$ when the distributions are no longer the same.

Now, we show the existence of instances for which our bounds are tight. We say an instance is *admissible* if and only if it satisfies all the statistics generated by the log data. The following proposition shows all the bounds defined in Theorem 7 are partially tight.

**Proposition 8** (Tightness of Transfer). *For any log with $u_{k,z}, l_{k,z}$ as defined in Theorem 7 the following statements hold:*
*1. There exists an admissible instance where upper bounds $\mu_{k,z} = u_{k,z}$, for all $k \in [K_0]$ and $z \in \mathcal{Z}$.*
*2. For each $k \in [K_0]$, there exists a (separate) admissible instance such that $\mu_{k,z} = l_{k,z}$ for each $z \in \mathcal{Z}$.*

The learning procedure is then carried out by using the bounds of Theorem 7 to instantiate GLUE (Algorithm 2, with regret as in Theorem 6) for each partial context $z \in \mathcal{Z}$ with $u_k = u_{k,z}$ and $l_k = l_{k,z}$.

## 6.2 Confounded Logs for Linear Bandits

**The Oracle Environment:** Consider a linear bandit environment with $K$ arms $a_k = (a_{k,z}, a_{k,u})$ for $i \in [K]$ and (random) latent vector $\theta^* = (\theta_z^*, T_u)$ such that $\theta_z^*$ is fixed. Suppose $a, \theta^* \in \mathbb{R}^p$, and for all $k \in [K]$, $a_{u,k}, T_u \in \mathbb{R}^d$ Further, we assume that $T_u$ is such that each entry of the vector is always between $[m, M]$, and $\|a_k\| = 1$ for all $k \in [K]$ (the equality can be generalized to bounds on the norm). At each round $t$, $T_u(t)$ is drawn independently from some distribution $\mathcal{C}$ and the full vector $\theta^*(t) = (\theta_z^*, T_u(t))$ is revealed to the oracle, based on which the arm $A_t = \arg\max_{k \in [K]} \langle a_k, \theta^*(t) \rangle$ is played. The oracle then observes the reward $Y_t = \langle A_t, \theta^*(t) \rangle + \eta_t$ where $\eta_t$ is the conditionally $\sigma(A_t)$ subgaussian bounded random variable with respect to the filtration generated by the reward observations up to time $t-1$ and arm choices up to time $t$.

**Confounded Logs:** We define $\hat{k}(T)$ to be the index of the best arm when the random realization of $T_u$ was $T$, i.e., $\hat{k}(T) = \arg\max_{k \in [K], T_u = T} \langle a_k, \theta^* \rangle$ . The oracle records the tuple $(\hat{k}(T_u(t)), \theta_z^*, Y_t)$ at the end of each round and omits the explicit information about $T_u(t)$. These tuples are collected over an infinite horizon to form the dataset $\mathcal{D} = \{(\hat{k}(T_u(t)), \theta_z^*, Y_t) : t \in \mathbb{N}\}$.

**The Agent Environment:** The agent is provided with the infinite log generated by the oracle and needs to interact with the same environment with no other information about $\theta^*$ being revealed at each time. Specifically, at each round $t$, the agent chooses an arm $A_t \in \{a_k : k \in [K]\}$ and observes $Y_t = \langle A_t, \theta^* \rangle + \eta_t$ with $T_u(t)$ being drawn independently according to $\mathcal{C}$ and $\eta_t$ the conditionally $\sigma(A_t)$-subgaussian noise as before. Since the latent vector $\theta^*$ is hidden and is random at each round, the agent seeks to minimize the average regret given by

$$R_T = \sum_{t=1}^{T} \max_{k \in [K]} \mathbb{E}[\langle a_k - A_t, \theta^* \rangle]$$

$$= \sum_{t=1}^{T} \max_{k \in [K]} \langle a_{k,z} - A_{t,z}, \theta_z^* \rangle + \mathbb{E}[\langle a_{k,u} - A_{t,u}, T_u \rangle]$$

$$:= \sum_{t=1}^{T} \max_{k \in [K]} \langle a_{k,z} - A_{t,z}, \theta_z^* \rangle + \langle a_{k,u} - A_{t,u}, \theta_u^* \rangle$$

In the above, $\theta_u^*$ us the average value of the vector $T_u$ under the distribution $\mathcal{C}$ and $A_{t,z}, A_{t,u}$ are the parts of the arm $A_t$ corresponding to parameters $\theta_z^*, \theta_u^*$ respectively. Note that the first term in the sum is known fully to the agent as $\theta_z^*$ is recorded in the oracle logs and thus, the learning problem of the agent now reduces to a Linear Bandit over $d$ dimensions (the dimension of $\theta_u^*$) using the 'pseudo-rewards' $Y_t' = Y_t - \langle A_t, \theta_z^* \rangle$ for the chosen arm $A_t$. The genie policy in this case is to always play the arm $k^* = \arg\max_{k \in [K]} \langle a_{k,z}, \theta_z^* \rangle + \langle a_{k,u}, \theta_u^* \rangle$. We note that this definition averages out the randomness in $T_u$ and in general can not be inferred by knowing $\hat{k}(T_u)$ from the logs.

Mapping these back to our running healthcare example, the seen context $\theta_z^*$ are the known effects of each intervention in $\{a_k : k \in [K]\}$ on the patient that have been established using medical trials, while $T_u$ represents the responses patient to the intervention based on biomarkers that have not been explored in the trials. Thus, $\theta_u^*$ is the average of these unknown effects over the population of patients.

Further, the following quantities can be inferred from the logs:

**1.** $p_k$: The probability that arm $k$ was best in the oracle environment defined as $p_k = \mathbb{P}_{\mathcal{C}}\left(\hat{k}(T_u(t)) = k\right)$.

**2.** $\nu_k$: The average reward obtained when arm $k$ was optimal. This can be written as $\nu_k = \mathbb{E}\left[Y_t | \hat{k}(T_u(t)) = k\right]$.

Using these quantities, the following result suggests the mean bounds that the agent can infer from the logs:

**Theorem 9.** *For all arms $a_k : k \in [K]$, let $p_k, \nu_k$ be as defined above and $\mathbb{1}_d$ be the vector of all 1's in $d$ dimensions. Then, we have that*

$$\mathbb{E}[\langle a_k, \theta^* \rangle] - \nu_k p_k - (1 - p_k)\langle a_{k,z}, \theta_z^* \rangle \in [md\langle \mathbb{1}_d, a_{k,u} \rangle, Md\langle \mathbb{1}_d, a_{k,u} \rangle].$$

Using these bounds, the agent can then employ R-OFUL (Algorithm 1) in order to minimize its regret. Contrary to the stochastic setting, due to the linearity of rewards, we need not make any assumptions on the suboptimality gaps in the oracle environment to provide these mean bounds.

### 6.3 Empirical Validation

**Confounded Logs for Stochastic MABs:**

We present a recommendation systems example where an agent is tasked with learning the best movie to recommend to users in an online manner: the agent observes the user's occupation at each round and solves an independent bandit instance for each occupation. To assist in its learning, logs collected from an oracle are provided to this agent. The oracle has access to more information about users and observes occupation, age and gender in order to recommend movies, but only records the occupation for each user. We assume that environments do not change between the oracle and the agent, i.e., users are drawn randomly from the same population in both cases.

For this set of experiments, we use the Movielens 1M dataset of Harper & Konstan (2015). This data set consists of 6040 users, from whom over 1 million ratings of 3952 movies are collected. Each user is associated with a gender, age, occupation and zip-code. In this work, we ignore the zip-code. The ratings (or rewards) lie in the interval $[1, 5]$. These are normalized to $[0, 1]$ through normal shifting and scaling. The reward matrix of size $6040 \times 3952$ is then completed using the SoftImpute algorithm from Mazumder et al. (2010). Post matrix completion, we delete all users who's total reward across all movies is below 1500. We also further cluster the age attribute to have 4 classes: $0 - 17, 18 - 25, 25 - 49, 50+$. We also combine the occupation attributes appropriately to have 8 classes namely: Student, Academic, Scientific, Office, Arts, Law, Retired and Others.

After the above, the Occupation attribute is treated as the visible one, age and gender are hidden. We form meta-users by averaging all the users according to the tuple of attributes (gender,age). The reward matrix is also modified to now have each row represent a particular (gender, age) realization (for example, ('M', 0-17)). These reward matrices are separately computed for each of the 8 occupations. We then sample a random collection of 15 movies for each occupation to be considered as arms. Theorem 7 is then employed in order to form the confounded logs for each occupation class. The realized upper and lower bounds after this process are shown in Figure 6 in the appendix. In Figure 4, we provide the online learning behavior for two of these instances.

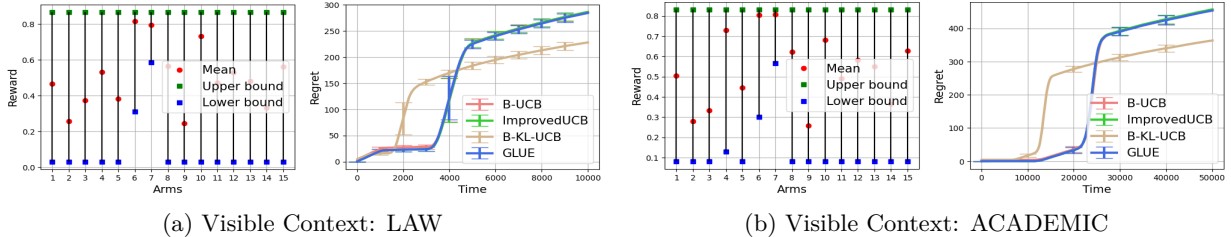

(a) Visible Context: LAW                    (b) Visible Context: ACADEMIC

Figure 4: Online Learning Behavior of two instances from the experiments using the Movielens 1M dataset. The visible context is displayed in the captions.

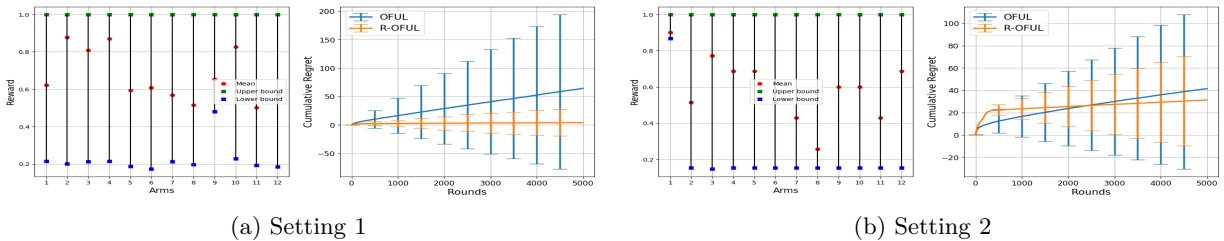

(a) Setting 1                              (b) Setting 2

Figure 5: Online Learning Behavior of two instances from the synthetic Linear Bandit setup. In each of the figures, the left figure summarizes the inferred bounds on $\langle \theta_u^*, a_k \rangle$ for each $k \in [12]$. The right figure displays the online experiment. Results are averaged over 200 independent runs and one standard deviation error bars are displayed. In the first setting, the bounds do not provide any improved subgaussian estimates, however, the restriction helps improve regret. In the second setting, the bounds provide improved exploration rates.

**Confounded Logs for Linear Bandits:** For this set of experiments, we use a synthetic setup: We first fix $\theta_z^*, \theta_u^* = (1, 2, 3, 4, 5)^T \in \mathbb{R}^5$. We normalize $\theta_z^*$ to have norm 1, and $\theta_u^*$ to have norm 0.9. For the arms, for each of the 12 arms, we draw $a_{k,z}$ to from a Folded Normal Distribution ($|X|$ is folded normal if $X$ is normally distributed) in $\mathbb{R}^5$. We set $a_{1,u}$ to be in a ball of radius 0.1 around $\theta_u^*$ at random and for all other $k$, $a_{k,u}$ is set to be a normalized 2-sparse vector in $\mathbb{R}^5$. With these, we generate logs by uniformly sampling $T_u(t)$ in a ball of radius 0.1 around $\theta_u^*$ independently for each $t$, then picking the best arm for this $T_u(t)$. Then we sample a noisy latent reward uniformly at random from a symmetric interval around $\langle T_u(t), a_{\hat{k}(T_u(t)),u} \rangle$, ensuring that it is in $[0, 1]$. The reward at each time is given by the sum of this noisy latent reward and $\langle \theta_z^*, a_{\hat{k}(T_u(t)),u} \rangle$. We collect logs of size $10^5$ to approximate the infinite logs.

We invoke Theorem 9 in order to infer upper and lower bounds on each of the 12 arms. Then, we spawn variants of the R-OFUL (Algorithm 1 and vanilla OFUL from Abbasi-Yadkori et al. (2011) and run an online linear bandit algorithm on our setting for 5000 iterations. The results are averaged over 200 independent runs. In Figure 5, we present out results. We see that when the bounds do no help in restricting the arms under consideration, R-OFUL matches vanilla OFUL (improvements here are from clipping the optimisitc indices for each arm). When non-trivial information can be gathered from the mean bounds, R-OFUL significatntly outperforms the vanilla variant.

# 7 Conclusion

In this work, we treated the problem of bandit learning with mean bounds in the linear and stochastic settings. Beginning with the study of how mean bounds lead to inference of sharp subgaussian factors when rewards are bounded, we present the R-OFUL algorithm for the linear setting and the GLUE algorithm for stochastic bandits that use these sharper factors to reduce exploration. In the linear case, by restricting the set of arms being considered at each round, we show that R-OFUL enjoys not only improved regret, but also increased compute efficiency in comparison to vanilla OFUL. In the stochastic setting, GLUE can offer comparable performance to the optimal B-KL-UCB algorithm when rich side information is available, and is never worse than the vanilla UCB policy. Further, we studied the practical use case of learning from (infinite) confounded logs where such mean bounds on rewards of actions can be inferred under mild assumptions on the latent environment.

Several avenues of future work were discussed over the course of exposition, chief among which is the question of lower bounds for the linear setting. While we present a compute-efficient baseline for the case of linear bandits with mean bounds, drawing from intuition in standard linear bandits, the lower bound is likely achieved by a complex, non-optimistic algorithm that further leverages the side information. Studying the optimality would also help characterize the environments in which the mean bounds improve regret performance. Further, the case of learning from finitely many confounded observations, which manifests as a problem of learning from probabilistic mean bounds remains open. In Corollary 6.1, we provide a natural a regret upper bound to the performance of GLUE in this case (which can also be extended to R-OFUL for linear bandits) which serves as a baseline. Finally, our assumption of bounded rewards led to tighter subgaussian factors and regret improvements. Identifying other use cases and modes of side information with arbitrary, potentially unbounded, reward distributions from which such improved factors can be computed also serves an an interesting direction.

### Acknowledgements

We sincerely thank all the reviewers and editors for the feedback that helped improve the quality of the paper. This research was partially supported by NSF Grants 1826320, 2019844, 2107037 and 2112471, ARO grant W911NF-17-1-0359, US DOD grant H98230-18-D-0007, ONR Grant N00014-19-1-2566 and the US DoT supported D-STOP Tier 1 University Transportation Center.

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

## A   Extreme bounds imply tighter subgaussian bounds

*Proof of Lemma 1.* Since $e^{-mx}$ and $e^{(1-m)x}$ are both convex, so is $f(x)$ (since $m, 1 - m \geq 0$). Thus, $f$ is minimized when

$$f'(x) = m(1-m)\left(e^{(1-m)x} - e^{-mx}\right) = 0$$
$$\iff e^{(1-m)x} = e^{-mx}$$
$$\iff (1-m)x = -mx$$
$$\iff x = 0$$

Therefore, the minimum value is $f(0) = 1$.

Now, we have

$$f_{1-m}(x) = (1-m)e^{(1-(1-m))x} + (1 - (1-m))e^{-(1-m)x} = (1-m)e^{-m(-x)} + me^{(1-m)(-x)} = f_m(-x).$$

Finally, the limit at $x = 0$ can be computed by using L'Hopital's rule. $\qquad\square$

We now prove each of the facts in Lemma 2 separately. Together, these two facts and Lemma 1, we have that there exist a $\sigma < 0.5$ for all $m \neq 0.5$.

### A.1   Non-existence of Positive Maximizer for Large Means

We begin with the case when $m > 0.5$ and prove some useful Lemmas that will help establish our result.

**Lemma 10.** *Let $A = mx + \frac{m(1-m)x^2}{2}$, $B = x + \log(m)$. Then, $A > B$ for all $m \in (0.5, 1)$ and $x \in \mathbb{R}$*

*Proof.* We have that

$$mx + \frac{m(1-m)x^2}{2} - (x + \log(m)) > 0 \iff m(1-m)x^2 + 2(m-1)x - 2\log(m) > 0$$

Call $m(1-m)x^2 + 2(m-1)x - 2\log(m) = h(x)$, differentiating and setting it to 0 gives $2m(1-m)x = 2(1-m) \implies x = \frac{1}{m}$. Thus, $\min_x h(x) = h\left(\frac{1}{m}\right) = \frac{m-1}{m} - 2\log(m)$. Since $\log(1+y) < y$ for all $y > -1$, with $m > 0.5$, we can write

$$\frac{m-1}{m} - 2\log(m) > \frac{m-1}{m} - 2(m-1) = \frac{(m-1)(1-2m)}{m} > 0.$$

$\qquad\square$

**Lemma 11.** *With $A$ and $B$ as defined in Lemma 10, the function $e^A - e^B$ is non-decreasing for $m \in (0.5, 1)$ and $x \in \left[\frac{1}{m}, \infty\right)$.*

*Proof.* We have that

$$e^A - e^B \text{ non-decreasing} \iff \frac{d(e^A - e^B)}{dx} \geq 0$$
$$\equiv \frac{dA}{dx}e^A - \frac{dB}{dx}e^B \geq 0$$
$$\equiv (m + m(1-m)x)\, e^A - e^B \geq 0.$$

From Lemma 10,

$$(m + m(1-m)x)\, e^A - e^B > e^B\, (m + m(1-m)x - 1)$$
$$\therefore e^A - e^B \text{ non-decreasing} \impliedby e^B\, (m + m(1-m)x - 1) \geq 0$$
$$\equiv m(1-m)x - (1-m) \geq 0 \equiv x \geq \frac{1}{m}.$$

$\qquad\square$

**Lemma 12.** *With $A$ and $B$ as defined in Lemma 10, the function $e^A - e^B$ is increasing for $m \in (0.5, 1)$ and $x \in \left(0, \frac{1}{m}\right)$.*

*Proof.* We begin with

$$e^A - e^B \text{increasing} \iff \frac{d(e^A - e^B)}{dx} > 0$$

$$\equiv \frac{dA}{dx}e^A - \frac{dB}{dx}e^B > 0$$

$$\equiv (m + m(1-m)x)\, e^A - e^B > 0$$

$$\equiv e^B \left((m + m(1-m)x)\, e^{A-B} - 1\right) > 0$$

$$\equiv (m + m(1-m)x)\, e^{A-B} > 1$$

$$\equiv e^{B-A} < (m + m(1-m)x)$$

Note that at $x = 0$, both sides of this equation compute to $m$. Thus, the inequality holds if $\frac{d(e^{B-A})}{dx} < \frac{d(m+m(1-m)x)}{dx}$ for $x > 0$. We have that $\frac{d(e^{B-A})}{dx} = e^{B-A} \cdot \frac{d(B-A)}{dx} = (1-m)(1-mx)e^{B-A}$ and $\frac{d(m+m(1-m)x)}{dx} = m(1-m)$. Thus, we require that $(1 - mx)e^{B-A} < m$. Let $k(x) = (1-mx)e^{B-A}$. Then, we have that

$$k'(x) = (1-mx)\frac{d\left(e^{B-A}\right)}{dx} - me^{B-A} = e^{B-A}\left((1-m)(1-mx)^2 - m\right).$$

Note that $(1-mx)^2$ decreases from 1 to 0 in $(0, \frac{1}{m})$. Therefore, $(1-mx)^2 < \max_{x \in (0, \frac{1}{m})}(1-mx)^2 = 1 < \frac{m}{1-m}$ for $m > 0.5$. Therefore, $(1-m)(1-mx)^2 - m < 0$ in $(0, \frac{1}{m})$. Since $e^{B-A} > 0$, $k(x)$ is also decreasing in this range.

Thus, we have that for $m \in (0, \frac{1}{m})$,

$$(1-mx)e^{B-A} < \max_{x \in (0, \frac{1}{m})}(1-mx)e^{B-A} = 1 - m.$$

$\square$

**Lemma 13.** *For all $m \in (0.5, 1)$ and $x > 0$, we have that $\log\left(f_m(x)\right) < \frac{m(1-m)x^2}{2}$.*

*Proof.* With $A, B$ as defined in Lemma 10, we require that

$$\log\left(f_m(x)\right) < \frac{m(1-m)x^2}{2}$$

$$\equiv me^x + 1 - m < \exp\left(mx + \frac{m(1-m)x^2}{2}\right) = e^A$$

$$\equiv 1 - m < e^A - me^x = e^A - e^{x+\log(m)} = e^A - e^B.$$

From Lemmas 11 and 12, we have that $e^A - e^B$ is non-decreasing in $(0, \infty)$. Thus, $e^A - e^B > \min_{x \in (0, \infty)} e^A - e^B = 1 - m$, giving us the result. $\square$

**Lemma 14** (Part 1 of Lemma 2)**.** *When $m \in (0.5, 1)$, the function $\frac{2}{x^2}\log\left(f_m(x)\right)$ is not maximized at any $x > 0$.*

*Proof.* Using 13, we have for any $m \in (0.5, 1), x > 0$,

$$\frac{2}{x^2}\log\left(f_m(x)\right) < \frac{2}{x^2} \cdot \frac{m(1-m)x^2}{2} = m(1-m).$$

However, Lemma 1 shows that $\lim_{x \to 0} \frac{2}{x^2}\log\left(f_m(x)\right) = m(1-m)$. Therefore, the function is not maximized when $x > 0$. $\square$

### A.2 Existence of Positive Maximizer for Small Means

We now move on to the case when $m < 0.5$. As before, we will establish the result using some useful lemmas.

**Lemma 15.** *Let $A = mx + \frac{m(1-m)x^2}{2}$ and $B = x + \log(m)$. Then, for $m \in (0, 0.5)$, we have that the function $e^A - e^B$ is*

  a) *decreasing for $x \in (0, x_1)$ where $x_1 = \frac{1}{m} - \frac{1}{\sqrt{m(1-m)}}$.*

  b) *increasing for $x > \frac{2}{m}$.*

*Proof.* **Part a:**
We require that in the given range,

$$\frac{dA}{dx}e^A - \frac{dB}{dx}e^B < 0$$
$$(m + m(1-m)x) < e^{B-A}.$$

Both sides of the above inequality compute to $m$ at $x = 0$. Therefore, the inequality holds for $x \in (0, x_1)$ if

$$\frac{d(m + m(1-m)x)}{dx} = m(1-m) < (1-m)(1-mx)e^{B-A} = \frac{d(e^{B-A})}{dx}$$
$$\text{That is, } m < (1-mx)e^{B-A}.$$

Let $k(x) = (1 - mx)e^{B-A}$. Then, we have $k'(x) = e^{B-A}\left((1-m)(1-mx)^2 - m\right) > 0 \iff \left((1-m)(1-mx)^2 - m\right) > 0$. Let $p(x) = (1-m)(1-mx)^2 - m$. Then, we have that $p(x) = 0$ at $x = \frac{1}{m} \pm \frac{1}{\sqrt{m(1-m)}}$. Therefore, $p(x) > 0$ for $x < x_1$.

Thus, $k(x)$ is increasing in $(0, x_1)$ and hence $k(x) > k(0) = m$. This implies that $(m + m(1-m)x) < e^{B-A}$ in $(0, x_1)$ which leads to the result.

**Part b.**
We first note that at $x = \frac{2}{m}$, $A = \frac{2}{m} > B = \frac{2}{m} + \log(m)$ since $m < 1$. Further, $\frac{dA}{dx} = m + m(m-1)x > 1 = \frac{dB}{dx}$ for all $x > \frac{1}{m}$. Thus, $A > B$ for $x > \frac{2}{m}$.

Thus, we get that

$$\frac{dA}{dx}e^A - \frac{dB}{dx}e^B = (m + m(1-m)x)\,e^A - e^B > e^B\,(m + m(1-m)x - 1) > 0 \qquad \text{for } x > \frac{1}{m}$$

And thus, $e^A - e^B$ is increasing in this range. $\qquad\square$

**Lemma 16** (Part 2 in lemma 2)**.** *For all $m \in (0, 0.5), x > 0$, we have that with $x_1$ as defined in Lemma 15*

  a) *For all $x \in (0, x_1)$, $\frac{2}{x^2}\log\left(f_m(x)\right) > m(1-m)$,*

  b) *For all $x > \frac{2}{m}$, $\frac{2}{x^2}\log\left(f_m(x)\right) < m(1-m)$.*

*Proof.* **Part a.**
For this, we require that in $(0, x_1)$,

$$\log\left(f_m(x)\right) > \frac{m(1-m)x^2}{2}$$
$$\iff e^{-mx}\left(me^x + 1 - m\right) > \exp\left(\frac{m(1-m)x^2}{2}\right)$$
$$\iff 1 - m > e^A - e^B$$

with $A, B$ as in Lemma 15. Using part a of this Lemma, the inequality holds since $e^A - e^B$ is decreasing in $(0, x_1)$ and is maximized at $x = 0$ with value $1 - m$.

**Part b.**
To see this, we require that

$$\log\left(f_m(x)\right) < \frac{m(1-m)x^2}{2} \iff 1 - m < e^A - e^B$$

This holds since $e^A - e^B$ is increasing when $x > \frac{2}{m}$ (Lemma 15 part b) and is hence minimized at $x = \frac{2}{m}$ with value $(1-m)e^{2/m} > (1-m)$ since $m < 1$. □

## A.3 Putting it all together

The following lemma will help us prove our final result.

**Lemma 17.** *Define $C = mx + \frac{x^2}{8}$ and $D = x + \log(m)$. Then, $e^C - e^D$ is increasing in $(0, \infty)$ for any $m \in (0, 0.5)$.*

*Proof.* This hold if and only if $\frac{dC}{dx}e^C - \frac{dD}{dx}e^D > 0$ for all $x > 0$. That is,

$$\left(m + \frac{x}{4}\right)e^C - e^D > 0 \iff m + \frac{x}{4} > exp(D - C).$$

For $x = 0$, both sides of this inequality compute to $m$. Thus, the inequality holds if

$$\frac{d}{dx}\left(m + \frac{x}{4}\right) > \frac{d(D-C)}{dx}\exp(D-C) \equiv \frac{1}{4} > \left(1 - m - \frac{x}{4}\right)\exp(D-C).$$

Since $\exp(D - C) > 0$ for all $x$, the inequality is trivially true for $x > (1 - m)$. To see that this is true for $x \in (0, 4(1-m))$, define $k(x) = \left(1 - m - \frac{x}{4}\right)\exp(D - C)$. Since $k'(x) = \left(\left(1 - m - \frac{x}{4}\right)^2 - \frac{1}{4}\right)\exp(D - C)$, we have that

$$k'(x) = 0 \iff \left(\left(1 - m - \frac{x}{4}\right)^2 - \frac{1}{4}\right) = 0 \iff x = 4(1-m) \pm 2.$$

Therefore, in $(0, 4(1-m))$, $k(x) = 0 \iff x = 4(1-m) - 2 = 2(1-2m)$. Further, since $1 - m - \frac{x}{4} > 0$ for $x < 4(1-m)$, $x = 2(1-2m)$ must be a maximizer of $k(x)$. We are now left to prove that $k(2(1-2m)) < \frac{1}{4}$ for any $m < \frac{1}{2}$. For this we have

$$k(2(1-2m)) = \left(1 - m - \frac{1-2m}{2}\right)\exp\left(2(1-2m) + \log(m) - 2m(1-2m) - \frac{(1-2m)^2}{2}\right)$$

$$= \frac{1}{2}\exp\left(\log(m) + (1-2m)\left(2 - 2m - \frac{1-2m}{2}\right)\right)$$

$$= \frac{m}{2}\exp\left(\frac{(1-2m)(3-2m)}{2}\right).$$

Let $q(x) = \frac{x}{2}\exp\left(\frac{(1-2m)(3-2m)}{2}\right)$. We have that

$$q'(x) = \exp\left(\frac{(1-2m)(3-2m)}{2}\right)\left(\frac{1}{2} + \frac{x}{4}(8x - 8)\right)$$

$$= \exp\left(\frac{(1-2m)(3-2m)}{2}\right)\left(\frac{4x^2 - 4x + 1}{2}\right)$$

$$= \frac{(1-2x)^2}{2} \exp\left(\frac{(1-2m)(3-2m)}{2}\right)$$
$$> 0 \quad \forall x$$

Thus, $q(x)$ is increasing for all $x$. Specifically, we have that for any $m \in (0, \frac{1}{2})$, $k(2(1-2m)) = q(m) < q(\frac{1}{2}) = \frac{1}{4}$. Thus, $k(2(1-2m)) < \frac{1}{4}$ for any $m \in (0, \frac{1}{2})$ and $x \in (0, 4(1-m))$. Therefore, $\frac{1}{4} > \left(1 - m - \frac{x}{4}\right) \exp(D-C)$ and thus, $e^C - e^D$ is increasing for all $x > 0$. $\qquad\square$

We are now ready to present our final result.

*Proof of Theorem 3.* **Subgaussianity**

Let $g_m(x) = \frac{2}{x^2} \log\left(f_m(x)\right)$. For $m < 0.5$, part b of Lemma 1 and Theorem 14 imply that the maximum of $g_m(x)$ is not achieved at any negative $x$. Further, Theorem 16 implies that there exists $x > 0 : g_m(x) > m(1-m)$. Additionally, since $g_m(x) < m(1-m)$ for $x > \frac{2}{m}$, we have that the function of interest is maximized at some $0 < x < \frac{2}{m}$.

Analogously, for $m > 0.5$, this function is maximized at some $\frac{-2}{m} < x < 0$.

Thus, for any random variable $X \in [0,1]$ with mean $m \in (0,1), m \neq 0.5$, we have $\mathbb{E}[\exp\left(s(X-m)\right)] \leq \exp\left(\frac{s^2\sigma^2}{2}\right)$. Thus, $X$ is $\sigma$-subgaussian.

**Bounds on $\sigma$**

From Part 2 of Lemma 2 (Lemma 16), we have that

$$\sigma^2 \geq \lim_{x \to 0} \frac{2}{x^2} \log(f_m(x)) = m(1-m).$$

Thus, we are left with showing that $\sigma < \frac{1}{2}$ when $m < 0.5$. Exploiting part b of Lemma 1 implies the result for $m < 0.5$.

We require that $\max_{x \in \mathbb{R}} \frac{2}{x^2} \log\left(f_m(x)\right) < \frac{1}{4}$. However, since we have already shown that this maximum is achieved at a positive $x$, it is sufficient to show $\max_{x>0} \frac{2}{x^2} \log\left(f_m(x)\right) < \frac{1}{4}$. That is, for all $x > 0$,

$$\frac{2}{x^2} \log\left(f_m(x)\right) < \frac{1}{4}$$
$$\equiv \log\left(f_m(x)\right) < \frac{x^2}{8}$$
$$\equiv me^x + 1 - m < \exp\left(mx + \frac{x^2}{8}\right)$$
$$\equiv 1 - m < \exp\left(mx + \frac{x^2}{8}\right) - \exp(x + \log(m))$$

From Lemma 17, we have that the RHS is increasing in $(0, \infty)$ and is thus minimized when $x = 0$ with minimum value $\frac{1}{4}$. This gives us the result. $\qquad\square$

## B  Linear Bandits with Mean Bounds

Now we concentrate on the setting of Section 4 and provide proofs for the result of Theorem 4. First, we will show that the tight bounds we propose in Equation 2 are valid and tight. Then, we prove that the best instantaneous arm at each time is always contained in the restricted set at each time. Next, we show that the quantity $\sigma_t$ is a valid upper bound on the subgaussian factor for all arms in the restricted set. This culminates in our final regret result. We begin with the bounds.

**Claim 1.** *The solutions to the optimization problems in Equation 2 is tight, that is, $\nexists \theta' \in \mathcal{C}_b$ such that for any $a \in \mathcal{A}$, $\langle a, \theta' \rangle < l_a$ or $\langle a, \theta' \rangle > u_a$*

*Proof.* First we note that the constraint set $\mathcal{C}_b$ is non-empty since $\theta^* \in \mathcal{C}_b$ and thus the quantities $l_a, u_a$ are well-defined for each $a \in \mathcal{A}$. Suppose $\exists \theta' \in \mathcal{C}_b$ with $\langle a, \theta' \rangle < l_a$ for some action $a$. This implies that $l_a \neq \min_{\theta \in \mathcal{C}_b} \langle a, \theta \rangle$ which is a contradiction. A similar argument for the upper bound completes the proof. $\quad\square$

Note that at this point, $\mathcal{C}_b$ is redefined to be the set $\{\theta : \|\theta\| \in [m, M], \forall a \in \mathcal{A}, \langle a, \theta \rangle \in [l_a, u_a]\}$. We now show that the best arm is always in contention at any time.

**Lemma 18.** *Let $a_t^* = \arg\max_{a \in \mathcal{A}_t} \langle a, \theta^* \rangle$ be the instantaneous best arm. Then, $a_t^* \in \mathcal{A}_r(t)$.*

*Proof.* First, we show that $a_t^* \in \mathcal{A}_p(t)$. Suppose that $u_{a_t^*} < l_{max}(t) \implies \langle a, \theta^* \rangle \leq u_{a_t^*} < l_{max}(t) \leq \langle \hat{a}_t, \theta^* \rangle$, that is, $\langle a_t^*, \theta^* \rangle < \langle \hat{a}_t, \theta^* \rangle$. This is a contradiction to the definition of $a_t^*$ and thus, $a_t^* \in \mathcal{A}_p(t)$.

To show that $a_t^* \in \mathcal{A}_{prune}(t)$, we first observe that

$$
\begin{aligned}
l_{max}(t) &\leq \langle \hat{a}_t, \theta^* \rangle \\
&= \|\hat{a}\| \|\theta^*\| \cos(\text{ang}(\hat{a}_t, \theta^*)) \\
&\leq M \cos(\text{ang}(\hat{a}_t, \theta^*)) \qquad\qquad \because \|\theta^*\| \leq M, \|a\| = 1 \\
\implies \text{ang}(\hat{a}_t, \theta^*) &\leq \cos^{-1}\left(\frac{l_{max}(t)}{M}\right) = \alpha_t.
\end{aligned}
$$

Further, since $l_{max}(t) \leq \langle a_t^*, \theta^* \rangle$, we get that $\text{ang}(a_t^*, \theta^*) \leq \alpha_t$. Combining these two, we get that $\text{ang}(\hat{a}_t, a_t^*) \leq 2\alpha_t$ and thus, $a_t^* \in \mathcal{A}_r(t)$. $\quad\square$

The following lemma proves that the quantity $\sigma_t$ is a valid upper bound to the s.g-factor of any arm in the restricted set.

**Lemma 19.** *For any arm $a \in \mathcal{A}_r(t)$, $\sigma(a) \leq \sigma_t$.*

*Proof.* From Corollary 3.1, we have that $\sigma(a) \leq \psi(l_a, u_a)$ for any arm $a \in \mathcal{A}_r(t)$. Therefore, it follows that $\sigma(a) \leq \max_{a' \in \mathcal{A}_r(t)} \psi(l_a', u_a')$.

We are only left to prove that $\sigma(a) \leq m\psi(3\alpha_t)$. For this, we note that the arm in $\mathcal{A}_r(t)$ with lowest mean reward is at most $3\alpha_t$ away from $\theta^*$. This happens when $\theta^*$ is at an angle $\alpha_t$ away from $\hat{a}_t$ and there exists some arm $a_{bad}$ at an angle $\alpha_t$ on the opposite side of $\hat{a}_t$. For this fictitious arm $a_{bad}$, we have $\langle a_{bad}, \theta^* \rangle = \|a_{bad}\| \|\theta^*\| \cos(3\alpha_t) \geq m \cos(3\alpha_t)$. Now, there are two cases:
1. If $m \cos(3\alpha_t) < 0.2178a + 0.7822b$: Then, $\psi(m \cos(3\alpha_t), b) = \frac{(b-a)^2}{4}$ and $\forall a \in \mathcal{A}_r(t), \sigma(a) \leq \psi(m \cos(3\alpha_t), b)$ follows by definition.
2. If $m \cos(3\alpha_t) \geq 0.2178a + 0.7822b$: Then any arm $a \in \mathcal{A}_r(t)$, $\langle a, \theta^* \rangle \geq \langle a_{bad}, \theta^* \rangle \geq m \cos(3\alpha_t)$. Using this as a lower bound for the arm $a$, we can write $\sigma(a) \leq \psi(m \cos(3\alpha_t), u_a) = \psi(m \cos(3\alpha_t), b)$.
This completes the proof. $\quad\square$

We now prove our final regret result.

*Proof of Theorem 4.* Let $S_t' = \sum_{n=1}^{t} \frac{\eta_n a_n}{\sigma_n}$, where the random variable $\eta_t$ is conditionally $\sigma_{a_t}$-subgaussian. From Lemma 19, $\sigma_{a_t} \leq \sigma_t$ and thus, $\eta_t$ is also conditionally $\sigma_t$-subgaussian.

We claim that $M_t(x) = \exp\left(\langle x, S_t' \rangle - \frac{\|x\|_{V_t}^2}{2}\right)$ with $M_0(x) := 1$ is a supermartingale with respect to the filtration $\mathcal{F}_{t-1} = \sigma(a_1, Y_1, ..., a_{t-1}, Y_{t-1}, a_t)$. To see this, observe that

$$
M_t(x) = \exp\left(\sum_{n=1}^{t}\left(\frac{\eta_n \langle x, a_n \rangle}{\sigma_n} + \frac{\|x\|_{a_n a_n^T}}{2}\right)\right)
$$

$$\implies \mathbb{E}\left[M_{t-1}|\mathcal{F}_{t-1}\right] = M_{t-1}(x) \cdot \mathbb{E}\left[\exp\left(\frac{\eta_n\langle x, a_t\rangle}{\sigma_t} + \frac{\|x\|_{a_t a_t^T}}{2}\right)\right]$$

$$\leq M_{t-1}(x) \qquad\qquad \because \eta_t \text{ is conditionally } \sigma_t\text{-s.g.}$$

Now let $H = \lambda I_d$ and $h \sim \mathcal{N}(0, H^{-1})$. With some linear algebra, we can write

$$\overline{M}_t := \int_{\mathbb{R}^d} M_t(x)dh(x) = \sqrt{\frac{\lambda^d}{\det \overline{V}_t}} \exp\left(\frac{\|S'_t\|_{\overline{V}_t^{-1}}}{2}\right)$$

Since $M_t(x)$ is a supermartingale, so is $\overline{M}_t$. Therefore, using the Maximal inequality for non-negative supermartingales, we get that

$$\delta \geq \mathbb{P}\left(\exists t : \|S'_t\|^2_{\overline{V}_t^{-1}} \geq 2\log\left(\frac{1}{\delta}\right) + \log\left(\frac{\det \overline{V}_t}{\lambda^d}\right)\right)$$

$$\geq \mathbb{P}\left(\exists t : \left\|\sum_{n=1}^t \eta_n a_n\right\|^2_{\overline{V}_t^{-1}} \geq \gamma^2\left(2\log\left(\frac{1}{\delta}\right) + \log\left(\frac{\det \overline{V}_t}{\lambda^d}\right)\right)\right) \tag{10}$$

The last step uses the fact that $\sum_{n=1}^t \frac{\eta_n a_n}{\sigma_t} \geq \frac{1}{\max_{n \leq t}\sigma_n}\sum_{n=1}^t \eta_n a_n = \frac{1}{\gamma}\sum_{n=1}^t \eta_n a_n$. Now, with $V_t = \sum_{n=1}^t a_n a_n^T$, we have

$$\left\|\hat{\theta}_t - \theta^*\right\|_{\overline{V}_t} = \left\|\overline{V}_t^{-1}\sum_{n=1}^t \eta_n a_n + \left(\overline{V}_t^{-1}V_t - I_d\right)\theta^*\right\|_{\overline{V}_t}$$

$$\leq \left\|\sum_{n=1}^t \eta_n a_n\right\|_{\overline{V}_t^{-1}} + \sqrt{\lambda\theta^{*T}\left(I - \overline{V}_t^{-1}V_t\right)\theta^*}$$

$$\leq \left\|\sum_{n=1}^t \eta_n a_n\right\|_{\overline{V}_t^{-1}} + \sqrt{\lambda}\|\theta^*\| \leq \left\|\sum_{n=1}^t \eta_n a_n\right\|_{\overline{V}_t^{-1}} + \sqrt{\lambda}M. \tag{11}$$

Combining Equations 10 and 11, we get that with probability at least $1 - \delta$,

$$\|\hat{\theta}_t - \theta^*\|_{\overline{V}_t^{-1}} \leq \sqrt{\lambda}M + \gamma\sqrt{2\log\left(\frac{1}{\delta}\right) + \log\left(\frac{\det \overline{V}_t}{\lambda^d}\right)}$$

$$\leq \beta_t(\delta) \qquad\qquad \because \frac{\det \overline{V}_t}{\lambda^d} \leq \left(\text{trace}\left(\frac{\overline{V}_t}{\lambda d}\right)\right)^d \leq \left(1 + \frac{t}{\lambda d}\right)^d$$

Now, with $\mathcal{E}_t = \left\{\theta : \|\hat{\theta}_t - \theta\|_{\overline{V}_t^{-1}} \leq \beta_t(\delta)\right\}$ observe that $\mathcal{C}_t = \mathcal{C}_b \cap \mathcal{E}_t$. Thus,

$$\mathbb{P}(\exists t : \theta^* \notin \mathcal{C}_t) \leq \mathbb{P}(\exists t : \theta^* \notin \mathcal{E}_t) \leq \delta.$$

The result of the theorem follows by using the standard arguments of Theorem 3 in Abbasi-Yadkori et al. (2011). □

## C GLUE for stochastic Multi-Armed Bandits with Mean Bounds

We begin with a few useful lemmas.

**Lemma 20.** *Any arm $k$ is $\sigma_k-$subgaussian. The best arm is always $\psi_k$-subgaussian.*

*Proof.* Both these are an application of Corollary 3.1. The first one follows using bounds $l_k, u_k$ for the suboptimal arm $k$. For the second, observe that $\mu^* \in [l_{max}, u_{k^*}]$, which are used to form $\psi_k$. $\qquad\square$

The next lemma proves that meta-pruned arms are only played a constant number of times asymptotically.

**Lemma 21.** *Let arm* $k \in K_1(\delta) := \{k \in [K] : \mu^* \geq u_k + \delta, k \neq 1\}$. *Then, for all* $n \geq 1$,

$$\mathbb{E}[T_k(n)] \leq \frac{5\psi_1^2}{(\max\{\delta, \mu^* - u_k\})^2}.$$

*Proof.* Since $k \in K_1(\delta) \implies \Delta_k > 0$, we have that $\{A_t = k\} \subseteq \{U_1(t) \leq u_k\}$. Thus,

$$\mathbb{E}[T_k(n)] = \mathbb{E}\left[\sum_{t=1}^{n} \mathbb{1}\{A_t = k\}\right]$$

$$\leq \mathbb{E}\left[\sum_{t=1}^{n} \mathbb{1}\{U_1(t-1) \leq u_k\}\right]$$

$$\leq \mathbb{E}\left[\sum_{t=1}^{n} \mathbb{1}\left\{\hat\mu_1(t) - \mu^* \leq u_k - \mu^* - \sqrt{\frac{2\psi_1^2 \log(f(t))}{T_1(t-1)}}\right\}\right]$$

$$\leq \sum_{t=1}^{n} \sum_{r=1}^{n} \exp\left(-\frac{r}{2\psi_1^2}\left(\sqrt{\frac{2\psi_1^2 \log(f(t))}{r}} + \mu^* - u_k\right)^2\right)$$

$$\text{(Using Union Bound, Lemma 20)}$$

$$\leq \sum_{t=1}^{n} \frac{1}{f(t)} \sum_{r=1}^{n} \exp\left(-\frac{r(\mu^* - u_k)^2}{2\psi_1^2}\right)$$

$$\leq \frac{2\psi_1^2}{(\mu^* - u_k)^2} \sum_{t=1}^{n} \frac{1}{f(t)}$$

$$\leq \frac{5\psi_1^2}{(\mu^* - u_k)^2}$$

$$\leq \frac{5\psi_1^2}{(\max\{\delta, \mu^* - u_k\})^2}$$

where the last inequality follows from the assumption on $u_k$. $\qquad\square$

The next lemma is a restatement of Lemma 8.2 in Lattimore & Szepesvári (2020) for standard subgaussian variables. It will be used to bound the number of plays of a suboptimal arm that is not meta-pruned.

**Lemma 22** (Lemma 8.2 in Lattimore & Szepesvári (2020)). *Let* $\{X_i\}$ *be a sequence of zero mean, independent* $\sigma$-*subgaussian random variables. Let* $\hat\mu_t = \frac{1}{t}\sum_{r=1}^{t} X_r, \delta > 0, a > 0, u = 2a\delta^{-2}$ *and*

$$\kappa = \sum_{t=1}^{n} \mathbb{1}\left\{\hat\mu_t + \sqrt{\frac{2a}{t}} \geq \delta\right\}$$

$$\kappa' = u + \sum_{t=\lceil u \rceil}^{n} \mathbb{1}\left\{\hat\mu_t + \sqrt{\frac{2a}{t}} \geq \delta\right\}$$

*Then,* $\mathbb{E}[\kappa] \leq \mathbb{E}[\kappa'] \leq 1 + 2\delta^{-2}\left(a + \sqrt{\sigma^2 \pi a} + \sigma^2\right)$ *for each* $n \geq 1$.

*Proof.* This is a restatement of the lemma for the general case of $\sigma$-subgaussian random variables.

Clearly, we have that $\mathbb{E}[\kappa] \le \mathbb{E}[\kappa']$. Thus,

$$
\begin{aligned}
\mathbb{E}[\kappa'] = \mathbb{E}\left[ u + \sum_{t=\lceil u \rceil}^{n} \mathbb{1}\left\{ \hat{\mu}_t + \sqrt{\frac{2a}{t}} \ge \delta \right\} \right] \\
= u + \sum_{t=\lceil u \rceil}^{n} \mathbb{P}\left( \hat{\mu}_t + \sqrt{\frac{2a}{t}} \ge \delta \right) \\
\le u + \sum_{t=\lceil u \rceil}^{n} \exp\left( -\frac{t}{2\sigma^2}\left( \delta - \sqrt{2a/t} \right)^2 \right) \qquad (X_i \sim \sigma - \text{subgaussian}) \\
\le 1 + u + \int_{u}^{\infty} \exp\left( -\frac{t}{2\sigma^2}\left( \delta - \sqrt{2a/t} \right)^2 \right) dt
\end{aligned}
$$

Now, using $x^2 = \frac{(\delta\sqrt{t} - \sqrt{2a})^2}{2\sigma^2}$, we have

$$
t = \frac{(\sqrt{2\sigma^2}x + \sqrt{2a})^2}{\delta^2} \implies dt = \frac{2\sqrt{2\sigma^2}}{\delta^2}\left( \sqrt{2\sigma^2}x + \sqrt{2a} \right) dx
$$

Note that the definition of $u = \frac{2a}{\delta^2}$ gives $t = u \implies x = 0$. Thus, we get

$$
\begin{aligned}
\mathbb{E}[\kappa'] &\le 1 + \frac{2a}{\epsilon^2} + \int_{0}^{\infty} \frac{2\sqrt{2\sigma^2}}{\delta^2} e^{-x^2}\left( \sqrt{2\sigma^2}x + \sqrt{2a} \right) dx \\
&\le 1 + \frac{2a}{\delta^2} + \frac{4\sigma^2}{\delta^2}\int_{0}^{\infty} x e^{-x^2} dx + \frac{4\sqrt{\sigma^2 a}}{\delta^2}\int_{0}^{\infty} e^{-x^2} dx \\
&= 1 + \frac{2a}{\delta^2} + \left( \frac{4\sigma^2}{\delta^2} \times \frac{1}{2} \right) + \left( \frac{4\sqrt{\sigma^2 a}}{\delta^2} \times \frac{\sqrt{\pi}}{2} \right) \\
&= 1 + \frac{2}{\delta^2}\left( a + \sqrt{\pi a \sigma^2} + \sigma^2 \right).
\end{aligned}
$$

Where we use that $\int_{0}^{\infty} x e^{-x^2} dx = \frac{1}{2}$ and $\int_{0}^{\infty} e^{-x^2} dx = \sqrt{\pi}/2$. $\qquad \square$

**Lemma 23.** *For the best arm, $\mathbb{E}\left[ \sum_{t=1}^{n} \mathbb{1}\{ U_1(t-1) \le \mu^* - \epsilon \} \right] \le \frac{5\psi_1^2}{\epsilon^2}$ for any $\epsilon > 0$.*

*Proof.* We have

$$
\begin{aligned}
\mathbb{E}\left[ \sum_{t=1}^{n} \mathbb{1}\{ U_1(t-1) \le \mu^* - \epsilon \} \right] &= \mathbb{E}\left[ \sum_{t=1}^{n} \mathbb{1}\{ U_1(t-1) \le \mu^* - \epsilon \} \right] \\
&= \mathbb{E}\left[ \sum_{t=1}^{n} \mathbb{1}\left\{ \hat{\mu}_1(t) - \mu^* \le -\epsilon - \sqrt{\frac{2\psi_1^2 \log(f(t))}{T_1(t-1)}} \right\} \right] \\
&\le \frac{5\psi_1^2}{\epsilon^2}
\end{aligned}
$$

This follows by using the union bound and Lemma 20 for Arm 1 (See Theorem 8.1 in Lattimore & Szepesvári (2020) for more details on the inequality). $\qquad \square$

**Lemma 24.** *If $k \in K_2(\delta) := \{ k \in [K] : \mu^* \le u_k + \delta, k \ne 1 \}$ for an arbitrary $\delta > 0$, then,*

$$
\mathbb{E}[T_k(n)] \le 1 + \frac{5\psi_1^2}{\Delta_k^2} + \frac{2}{\Delta_k^2}\left( \psi_k^2 \log(f(n)) + \sqrt{\psi_k^2 \sigma_k^2 \pi \log(f(n))} + \sigma_k^2 \right) + h(n).
$$

*Where $h(n) = \Omega\left( \log(f(n))^{2/3} \right)$.*

*Proof.* Since $k \neq 1$, $\{A_t = k\} \subseteq \{U_1(t-1) \leq \mu^* - \epsilon\} \cup \{U_k(t-1) > \mu^* - \epsilon, A_t = k\}$ for any $\epsilon \in (0, \Delta_k)$. Thus, we have that

$$
\mathbb{E}[T_k(n)] \leq \mathbb{E}\left[\sum_{t=1}^{n} \mathbb{1}\{U_1(t-1) \leq \mu^* - \epsilon\}\right] + \mathbb{E}\left[\sum_{t=1}^{n} \mathbb{1}\{U_k(t-1) > \mu^* - \epsilon, A_t = k\}\right]
$$

$$
\leq \frac{5\psi_1^2}{\epsilon^2} + \mathbb{E}[\sum_{t=1}^{n} \mathbb{1}\{U_k(t-1) > \mu^* - \epsilon, A_t = k\}] \qquad \text{(Using Lemma 23)}
$$

For the second term, following the steps in the Proof of Theorem 8.1 in Lattimore & Szepesvári (2020) and using Lemma 22 above, we get

$$
\mathbb{E}[\sum_{t=1}^{n} \mathbb{1}\{U_k(t-1) > \mu^* - \epsilon, A_t = k\}]
$$

$$
= \mathbb{E}\left[\sum_{t=1}^{n} \mathbb{1}\left\{\hat{\mu}_k(t-1) + \sqrt{\frac{2\psi_k^2 \log(f(t))}{T_k(t-1)}} > \mu^* - \epsilon, A_t = k\right\}\right]
$$

$$
\leq \mathbb{E}\left[\sum_{r=1}^{n} \mathbb{1}\left\{\hat{\mu}_{kr} - \mu_k > \Delta_k - \epsilon - \sqrt{\frac{2\psi_k^2 \log(f(n))}{r}}\right\}\right]
$$

$$
\leq 1 + \frac{2}{(\Delta_k - \epsilon)^2}\left(\psi_k^2 \log(f(n)) + \sqrt{\psi_k^2 \sigma_k^2 \pi \log(f(n))} + \sigma_k^2\right)
$$

Here, $\hat{\mu}_{kt}$ is the empirical mean of $t$ i.i.d samples from arm $k$. The last inequality follows from Lemma 22 with $a = \psi_k^2 \log(f(n))$, $\delta = (\Delta_k - \epsilon)$ and $\sigma = \sigma_k$ (using Lemma 20). Substituting this into the original expression, we get

$$
\mathbb{E}[T_k(n)] \leq \frac{5\psi_1^2}{\epsilon^2} + 1 + \frac{2}{(\Delta_k - \epsilon)^2}\left(\psi_k^2 \log(f(n)) + \sqrt{\psi_k^2 \sigma_k^2 \pi \log(f(n))} + \sigma_k^2\right)
$$

$$
\leq \inf_{\epsilon \in (0, \Delta_k)} \frac{5\psi_1^2}{\epsilon^2} + 1 + \frac{2}{(\Delta_k - \epsilon)^2}\left(\psi_k^2 \log(f(n)) + \sqrt{\psi_k^2 \sigma_k^2 \pi \log(f(n))} + \sigma_k^2\right) \qquad (12)
$$

Now, let $g(n) = \left(\psi_k^2 \log(f(n)) + \sqrt{\psi_k^2 \sigma_k^2 \pi \log(f(n))} + \sigma_k^2\right)$, and $0 < \epsilon = \frac{\Delta_k}{\alpha g(n)^{1/3} + 1} < \Delta_k$ for $\alpha = (2/5\psi_1^2)^{1/3}$. We have the following:

$$
\inf_{\epsilon \in (0, \Delta_k)} \frac{5\psi_1^2}{\epsilon^2} + 1 + \frac{2}{(\Delta_k - \epsilon)^2} g(n)
$$

$$
\leq 1 + \frac{5\psi_1^2}{\Delta_k^2}(\alpha g(n)^{1/3} + 1)^2 + \frac{2}{\Delta_k^2}\frac{(\alpha g(n)^{1/3} + 1)^2}{\alpha^2 g(n)^{2/3}} g(n)
$$

$$
= 1 + \frac{5\psi_1^2}{\Delta_k^2}\left(\alpha^2 g(n)^{2/3} + 2\alpha g(n)^{1/3} + 1\right) + \frac{2g(n)}{\Delta_k^2}\frac{(\alpha g(n)^{1/3} + 1)^2}{\alpha^2 g(n)^{2/3}}
$$

$$
= 1 + \frac{5\psi_1^2}{\Delta_k^2} + \frac{2g(n)}{\Delta_k^2} + \frac{g(n)^{2/3}}{\Delta_k^2}\left(5\psi_1^2 \alpha^2 + \frac{4}{\alpha}\right) + \frac{g(n)^{1/3}}{\Delta_k^2}\left(10\psi_1^2 \alpha + \frac{2}{\alpha^2}\right)
$$

$$
= 1 + \frac{5\psi_1^2}{\Delta_k^2} + \frac{2g(n)}{\Delta_k^2} + \underbrace{\frac{(20\psi_1^2)^{1/3}g(n)^{2/3}}{\Delta_k^2}\left(1 + (20\psi_1^2)^{1/3}\right) + \frac{3g(n)^{1/3}}{\Delta_k^2}\left(5\sqrt{2}\psi_1^2\right)^{2/3}}_{=h(n)}
$$

The result follows with $h(n)$ being defined as the last two terms of the expression above. $\qquad \square$

We are now ready to prove the theorem.

*Proof of Theorem 6.* The theorem now follows immediately using Lemmas 21 and 24. This is because we can decompose regret as

$$R_n = \sum_{t=1}^{n} \mathbb{E}[\mu^* - Y_t] = \sum_{K_1(\delta)} \Delta_k \mathbb{E}[T_k(n)] + \sum_{K_2(\delta)} \Delta_k \mathbb{E}[T_k(n)],$$

where $K_1(\delta) = \{k \in [K] : \mu^* > u_k + \delta, k > 1\}$ and $K_2(\delta) = \{k \in [K] : \mu^* \le u_k + \delta, k > 1\}$, for any $\delta > 0$.

The first part of the theorem follows by bounding the two summations using Lemmas 21 and 24 respectively. To prove the asymptotic part, we simply take $\delta \to 0$ and $n \to \infty$ in the above expression. $\square$

## D  OSSB for MABs with mean bounds and Bernoulli Rewards

Here, we derive closed form expressions for the semi-infinite optimization problem in Combes et al. (2017) in the case of Bernoulli rewards and bandits with mean bounds. In this section, we follow the notation in their paper. An instance $\theta$ is represented by a vector of arm means $(\theta_1, \theta_2, ... \theta_K)$. We say an instance $\theta$ is feasible if for each $k \in [K]$, we have that $\theta_k \in [l_k, u_k]$. Let $\Theta$ be the set of all feasible instances. With $\Theta$ as above, $\nu(\theta_k) \sim Bernoulli(\theta_k)$ and the mapping $k, \theta \mapsto \mu(k, \theta)$ as $\mu(k, \theta) = \theta_k$, our setting of Bernoulli bandits with mean bounds can hence be viewed as a Structured Bandit.

Define for any $\alpha, \beta \in \Theta$, $D(\alpha, \beta, k) = d(\alpha_k, \beta_k)$ where $d(\cdot, \cdot)$ is the Bernoulli kl-divergence function. Let $\mu^*(\theta) = max_{k \in [K]} \mu(k, \theta)$. Then, the optimization problem that determines the regret lower bound in our case can be given as:

$$\min_{\eta(k) \ge 0} \sum_{k \in [K]} \eta(k) \left(\mu^*(\theta) - \mu(k, \theta)\right)$$

$$s.t. \sum_{k \in [K]} \eta(k) D(\theta, \lambda, k) \ge 1 \quad \forall \lambda \in \Lambda(\theta)$$

$$\Lambda(\theta) = \{\lambda \in \Theta : D(\theta, \lambda, k_\theta^*) = 0, k^*(\theta) \neq k^*(\lambda)\}$$

We now present the solution of the above optimization for a given instance $\theta$.

For simplicity, consider the case with 2 arms: Arm 1 and Arm k. Let the instance $\theta \in \Theta$ be such that $\mu(1, \theta) = \mu^*(\theta)$. We now have two cases:

1. $\mu^*(\theta) > u_k$: In this case, the set $\Lambda(\theta)$ is empty since there is can be no other instance $\lambda \in \Theta$ with $\mu(1, \lambda) = \mu(1, \theta) = \mu^*(\theta)$ and Arm $k$ as optimal (since $\mu(k, \theta) \le u_k < \mu^*(\theta)$). Thus, the optimal solution to the optimization problem is $\eta^* = (0, 0)$.

2. $\mu_\theta^* \le u_k(\theta)$: In this case, we note that to satisfy the constraint over all instances in $\Lambda(\theta)$, it it sufficient to set $\eta(k) = \frac{1}{\min_{\mu \in [\mu^*(\theta), u_k]} d(\mu(k,\theta),\mu)} = \frac{1}{d(\mu(k,\theta),\mu^*(\theta))}$. This uses the fact that $d(a, x)$ as a function of $x$ is increasing in $[a, 1]$. Thus, the optimal solution in this case is $\eta^* = \left(0, \frac{1}{d(\mu(k,\theta),\mu^*(\theta))}\right)$.

Generalizing this to the case with $K$ arms, we get that the solution to this problem for a given $\theta \in \Theta$ is given by

$$\eta^*(k) = \begin{cases} 0 & \text{if } \mu^*(\theta) > u_k \text{ or if } k = k^*(\theta) \\ \frac{1}{d(\mu(k,\theta),\mu^*(\theta))} & \text{otherwise} \end{cases}$$

Finally, using this as the optima of the problem above, we have that the value at this optima is given by

$$C(\theta) = \sum_{k \in K_2(\theta)} \frac{\mu^*(\theta) - \mu(k, \theta)}{d\left(\mu(k, \theta), \mu^*(\theta)\right)} = \sum_{k \in K_2(\theta)} \frac{\Delta_k(\theta)}{d\left(\mu(k, \theta), \mu^*(\theta)\right)}$$

Thus, the asymptotic regret of OSSB reduces to $C(\theta)$ as $\epsilon, \gamma \to 0$. We note that in our case the OSSB algorithm uses the solution to the above optimization at the $t$-th round with $\theta$ replaced with the truncated empirical means, i.e. $\{\min(u_k, \max(l_k, \hat{\mu}_k(t))) \;\; \forall k \in [K]\}$.

# E    Confounded Logs

## E.1    Stochastic Multi-armed Bandit Setting

We begin with the proof of Theorem 7

*Proof of Theorem 7.* For the upper bound, we split the expectation into two parts: $(z, u)$ where $k$ is optimal, and $(z, u)$ where $k$ is not optimal. We have

$$
\begin{aligned}
\mu_{k,z} &= \sum_{u \in \mathcal{U}} \mu_{k,z,u} \mathbb{P}(u|z) \\
&= \sum_{u \in \mathcal{U}: k = k_{z,u}^*} \mu_{z,u}^* \mathbb{P}(u|z) + \sum_{u \in \mathcal{U}: k \neq k_{z,u}^*} \mu_{k,z,u} \mathbb{P}(u|z) \\
&\qquad \text{(Splitting the sum into parts based on the optimality of } k) \\
&\leq \sum_{u \in \mathcal{U}: k = k_{z,u}^*} \mu_{z,u}^* \mathbb{P}(u|z) + \sum_{u \in \mathcal{U}: k \neq k_{z,u}^*} \left( \mu_{z,u}^* - \underline{\delta}_z \right) \mathbb{P}(u|z) \\
&\qquad \text{(If } k \neq k_{z,u}^*, \text{ then its reward is at most } \mu_{z,u}^* - \underline{\delta}_z) \\
&= \sum_{u \in \mathcal{U}} \mu_{z,u}^* \mathbb{P}(u|z) - \underline{\delta}_Z \left( \sum_{u \in \mathcal{U}: k \neq k_{z,u}^*} \mathbb{P}(u|z) \right) \\
&= \mu_z - \underline{\delta}_Z (1 - p_z(k)) \qquad \text{(Using the definitions of } \mu_z \text{ and } p_z(k))
\end{aligned}
$$

The inequality follows since the logs are assumed to be collected under an optimal policy. This completes the proof of the upper bound.

To prove the lower bound, we fix an arbitrary $k \in [K_0]$ and $z \in \mathcal{Z}$. Recall that $K_>(k, z) = \{k' : k' \neq k, \mu_z(k') > \bar{\delta}_z p_z(k')\}$ and let $K_\leq(k, z) := [K_0] \backslash K_>(k, z) = \{k' : k' \neq k, \mu_z(k') \leq \bar{\delta}_z p_z(k')\}$. We note that these sets can be identified from the logged data because $\mu_z(k')$ and $p_z(k')$ can be derived for any $k'$ and $z$. Now we define the sets $\mathcal{U}_>(k, z), \mathcal{U}_\leq(k, z)$ as $\mathcal{U}_>(k, z) = \{u \in \mathcal{U} : k_{z,u}^* \in K_>(k, z)\}, \mathcal{U}_\leq(k, z) = \{u \in \mathcal{U} : k_{z,u}^* \in K_\leq(k, z)\}$

We now expand the mean of the arm $k$ under partial context $z$ as follows.

$$
\begin{aligned}
\mu_{k,z} &= \sum_{u \in \mathcal{U}: k = k_{z,u}^*} \mu_{z,u}^* \mathbb{P}(u|z) + \sum_{u \in \mathcal{U}: k \neq k_{z,u}^*} \mu_{k,z,u} \mathbb{P}(u|z) \\
&= \mu_z(k) + \sum_{u \in \mathcal{U}_\leq(k,z)} \mu_{k,z,u} \mathbb{P}(u|z) + \sum_{u \in \mathcal{U}_>(k,z)} \mu_{k,z,u} \mathbb{P}(u|z) \\
&\geq \mu_z(k) + \sum_{u \in \mathcal{U}_>(k,z)} (\mu_{z,u}^* - \bar{\delta}_z) \mathbb{P}(u|z) \\
&= \mu_z(k) + \sum_{k' \in K_>(k,z)} \mu_z(k') - \bar{\delta}_z \sum_{k' \in K_>(k,z)} p_z(k').
\end{aligned}
$$

The inequality holds because, firstly, we have $\mu_{k,z,u} \geq 0$ which we use for $u \in \mathcal{U}_\leq(k, z)$. Secondly, we have $\mu_{k,z,u} \geq (\mu_{z,u}^* - \bar{\delta}_z)$ by definition of $\bar{\delta}_z$, which we use for $u \in \mathcal{U}_>(k, z)$. Since this holds for any arbitrary $k, z$, this proves the lower bound. $\qquad \square$

Now we move on to the proof of our tightness claim

*Proof of 8.* **Upper Bound Tightness:**

For the tightness of the upper bound, notice that

$$
u_{k,z} = \sum_{u \in \mathcal{U}} \mu_{z,u}^* \mathbb{P}(u|z) - \underline{\delta}_z \sum_{u \in \mathcal{U}: k \neq k_{z,u}^*} \mathbb{P}(u|z)
$$

$$= \sum_{u \in \mathcal{U}: k = k^*_{z,u}} \mu_{k,z,u} \mathbb{P}(u|z) + \sum_{u \in \mathcal{U}: k \neq k^*_{z,u}} (\mu^*_{z,u} - \underline{\delta}_z) \mathbb{P}(u|z).$$

Which is the same as the quantity $\mu_{k,z} = \sum_{u \in \mathcal{U}} \mu_{k,z,u} \mathbb{P}(u|z)$ when

$$\mu_{k,z,u} = \mu^*_{z,u} - \underline{\delta}_z \quad \forall u \in \mathcal{U} : k \neq k^*_{z,u}$$

If the above equality holds for all $k \in [K_0]$ and all $z \in \mathcal{Z}$, we have that $\mu_{k,z} = u_{k,z}$. This instance is admissible since it maintains the means of the best arms are unchanged, and thus do not affect the quantities $\mu_z$ and $p_z(k)$ for any $k$. Thus, there exists an admissible instance where the upper bounds for each arm is tight.

**Lower Bound Tightness:**

Fix an arm $k$, and a partial context $z$. For the tightness of the lower bound for this particular arm $k$ and partial context $z$, we present an instance now. We assign $\mu_{k,z,u} = 0$ for all $u \in \mathcal{U}_{\leq}(k, z)$, and $\mu_{k,z,u} = \mu^*_{z,u} - \bar{\delta}_z$ for all $u \in \mathcal{U}_{\leq}(k, z)$. For this particular choice, it is easy to see that the above lower bound is tight. We now need to prove that this instance is admissible.

1. By construction we know $\mu_{k,z,u} \geq 0$ as $\mu^*_{z,u} \geq \bar{\delta}_z$ for all partial contexts $u \in \mathcal{U}_{>}(k, z)$. That $\mu_{k,z,u} \leq 1$, is easy to check.

2. We know that $(\mu^*_{z,u} - \mu_{k,z,u}) \leq \bar{\delta}_z$ by construction for all partial contexts $u \in \mathcal{U}_{>}(k, z)$, and due to the fact that $\mu^*_{z,u} \leq \bar{\delta}_z$ and $\mu_{k,z,u} = 0$ for all partial contexts $u \in \mathcal{U}_{\leq}(k, z)$.

3. We know that $(\mu^*_{z,u} - \mu_{k,z,u}) \geq \underline{\delta}_z$ by construction for all partial contexts $u \in \mathcal{U}_{>}(k, z)$, and due to the fact that $\mu^*_{z,u} \geq \underline{\delta}_z$ and $\mu_{k,z,u} = 0$ for all partial contexts $u \in \mathcal{U}_{\leq}(k, z)$. Here, $\mu^*_{z,u} \geq \underline{\delta}_z$ due to non-negativity of the mean-rewards and by definition of $\underline{\delta}_z$ (the smallest gaps).

4. Finally, $k$ is never optimal for any partial contexts $u \in \mathcal{U}_{\leq}(k, z) \cup \mathcal{U}_{>}(k, z)$. Indeed, for all the partial contexts $u \in \mathcal{U}_{\leq}(k, z)$, we have $\mu_{k,z,u} = 0$ and $\mu^*_{z,u} \geq \underline{\delta}_z > 0$. For all the partial contexts $u \in \mathcal{U}_{>}(k, z)$ we have $\mu_{k,z,u} = (\mu^*_{z,u} - \bar{\delta}_z) < \mu^*_{z,u}$. Therefore, for this instance we never observe $k$ for any partial contexts $u \in \mathcal{U}_{\leq}(k, z) \cup \mathcal{U}_{>}(k, z)$, which ensures the log statistics does not change.

This concludes that the instance is admissible, and it proves that the lower bound is tight. $\qquad\square$

## E.2 Confounded Logs for Linear Bandits with Fixed Arms

We now prove our result for mean bounds from confounded logs in the linear bandit case.

*Proof of Theorem 9.* Let $\mathcal{E}_k = \left\{ k = \hat{k}(T_u) \right\}$ be the event that arm $k$ was optimal under the full context $\theta^*$ when $T_u$ was drawn according to the distribution $\mathcal{C}$. By definition, we have $p_k = \mathbb{P}(\mathcal{E}_k)$ and

$$\begin{aligned}
\mathbb{E}[\langle a_k, \theta^* \rangle] &= \langle a_{k,z}, \theta^*_z \rangle + \mathbb{E}[\langle a_{k,u}, T_u \rangle] \\
&= \langle a_{k,z}, \theta^*_z \rangle + p_k \mathbb{E}\left[\langle a_{k,u}, T_u \rangle | \mathcal{E}_k\right] + (1 - p_k)\mathbb{E}\left[\langle a_{k,u}, T_u \rangle | \mathcal{E}^C_k\right] \\
&= p_k \nu_k + (1 - p_k)\left(\langle a_{k,z}, \theta^*_z \rangle + \mathbb{E}\left[\langle a_{k,u}, T_u \rangle | \mathcal{E}^C_k\right]\right)
\end{aligned}$$

Since $T_u$ is such that the values of each of its entries are in $[m, M]$, we can write $dm\langle \mathbb{1}_d, a_{k,u} \rangle \leq \mathbb{E}\left[\langle a_{k,u}, T_u \rangle | \mathcal{E}^C_k\right] \leq dM\langle \mathbb{1}_d, a_{k,u} \rangle$. Rearranging the above equation then gives us the required result. $\qquad\square$

## E.3 More Empirical Results

In Figure 6, we present the instances extracted out of each of the visible contexts of Occupation. The bounds are calculated as suggested by Theorem 7.

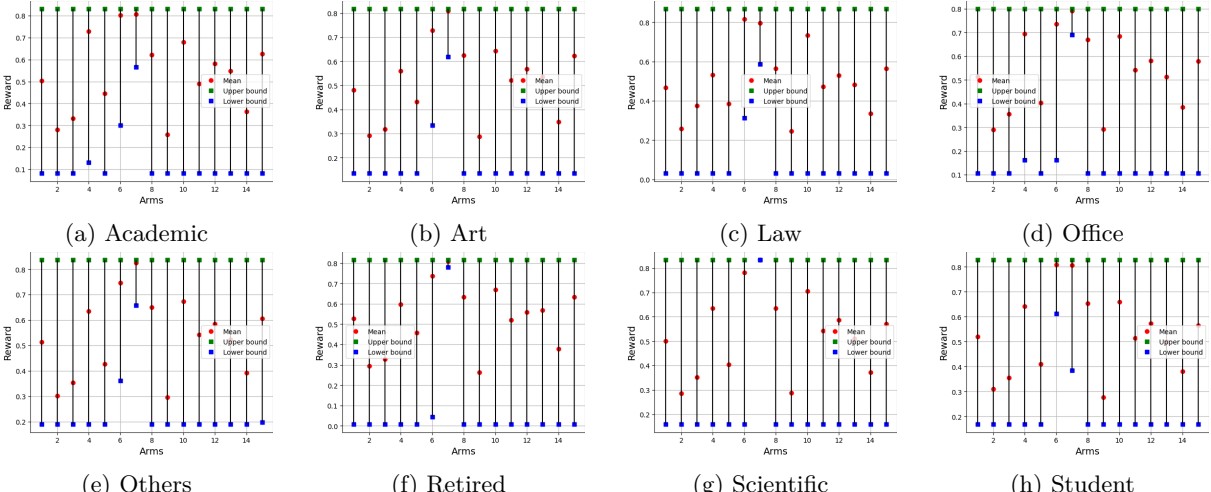

Figure 6: **Instances with Occupation as visible context:** The occupations are listed in the caption. The data filtration and movie selection process is explained in Section 6.3.

