# OpenReview forum: "Bandits with Mean Bounds"
_TMLR — Accepted by TMLR_

### Review · Reviewer_kenH · 2024-08-04

**Summary Of Contributions:**

​The paper addresses stochastic multi-armed bandit (MAB) and linear bandit problems incorporating side information, where the side information refers to additional bounds on the mean return of each arm. It introduces two natural algorithms for MAB and linear bandits setting and derives the corresponding regret bounds.

**Audience:**

No

**Broader Impact Concerns:**

Na.

**Claims And Evidence:**

No

**Requested Changes:**

See above weaknesses.

**Strengths And Weaknesses:**

Strengths:
The paper is well-written and easy to follow. The formulation is new to the existing literature.

Weaknesses:
My main concern is the technical contribution of the paper:

1. The main technical novelty of the paper is Section 3. All the derivations in the following sections are plug-ins of the results in Section 3 to the existing proofs/analyses. I don't see much challenge in such plug-ins. Coming back to Section 3, I think the result here is something people are well aware of, and the derivations are straightforward and don't involve advanced analysis.

2. The bound of the paper is only effective when the interval of the mean doesn't include 0.5 (as indicated in Corollary 3.1). This doesn't change the nature of the existing UCB, and in my view, its advantage over UCB is not essential. To see this, after the pruning step (which is trivial given this additional side information), the remaining arms have a lower bound larger than some constant, say, $l$, and an upper bound smaller than some constant, say $u$. For UCB algorithm, having all the mean rewards between $[l,u]$ is equivalent having all the mean rewards between $[0,1]$. We can't criticize the existing algorithms for them assuming the reward in $[0,1]$ but not $[l,u]$.

3. As I mentioned in the first point, I don't get the technical challenge when combining the new results in Section 3 with the existing analysis of UCB for MAB and linUCB. This limits the technical contribution of the work.

---

> ### Author Response · Authors · 2024-09-14
> **Response to Reviewer Comments**
>
> Thank you for your review of our paper. We would like to clarify some points that you have raised:
>
> The standard practice in literature is to use a subgaussian factor of $\frac{1}{2}$ for Bernoulli random variables as is the case in vanilla UCB algorithms. While the definition of the subgaussian factor is standard and the analysis is algebraic, inferring tighter subgaussian constants from mean bounds and using them in online decision making has not been explored before.
>
> Further, their application to the linear and stochastic MAB settings follow the standard sequence of arguments to establish regret upper bounds. However, we believe that the R-OFUL and GLUE algorithms we present and analyze are not trivial modifications to OFUL and UCB, respectively. Our major improvements in the linear bandit case comes from the restriction of arms, and in the stochastic MAB setting, from the sharing of information between arms using $\psi_k$. These are based on observations about the structure of the environments, which we believe are not trivial.
>
> In particular, in the linear case, the natural modification to OFUL would just be to apply the results of Section 3 to each arm. This modification does not lead to reduction in computation as all arms are still considered (after pruning). Additionally, the regret of this modification will scale with respect to the largest subgaussian factor of any of these arms. By restricting the set of arms as in R-OFUL, not only do we improve the computation complexity, but also (potentially) the largest subgaussian factors of arms in contention. Thus, our R-OFUL algorithm is superior (in regret and in computation cost) to the modification above. For the stochastic case, a simple application of results in Section 3 leads to ImprovedUCB that we describe and compare against in Section 5.3 and Figure 3. This modification employs improved subgaussian factors for all arms and also clips the respective UCB indices. However, as it does not share information among arms, GLUE offers improved regret performance when informative mean bounds are available. This leads to performance that is comparable to the optimal B-KL-UCB when the maximum lower bound is high (Figure 3a). In the same setting, the regret of ImprovedUCB is comparable to B-UCB.
>
> We would like to clarify that our methods lead to improvement in regret over standard UCB if there exist *at least one arm* which does not have 0.5 included in its mean bound. We refer to Figures 3a and 3d for examples. Indeed, the worst-case occurs when all mean bounds include 0.5, as in Figure 3b. In this case, GLUE matches the performance of B-UCB (vanilla UCB with clipped UCB indices). We note that in this case, the mean bounds are only effective in meta-pruning of arms, leading to some potential regret benefits over vanilla UCB. Further, as in Figure 3b, if the best arm has mean 0.5, the performance of the optimal B-KL-UCB algorithm matches that of vanilla UCB and GLUE. Our criticism of the naive learning algorithms is in their lack of ability to use useful information when available, and not their performance in the worst-case.
>
> Apart from the bandit setting, our use-case of Confounded logs that could also be of independent interest. It also presents a practical setting where our online approaches are naturally applicable.

---

> > ### Comment · Reviewer_kenH · 2024-09-16
> >
> > Thanks the authors for the clarifications and the updated manuscript. It does make the point clearer.
> >
> > And I understand the point that the algorithm is different from the existing UCB algorithms in that it involves a restriction of the arms. However, this difference is not a technical challenge in my view, and it is a natural thing to do when we are equipped with all the confidence intervals (to remove those arms whose UCB is smaller than some other arm's LCB). The existing elimination-based algorithms all follow this spirit.

---

> > > ### Author Response · Authors · 2024-09-16
> > > **Response to Comment**
> > >
> > > Thank you for your comments on our response. We would like to clarify that we go beyond the intuition of clipping arm indices in both the linear and stochastic MAB settings.  Improvements from restriction of arms based on LCB's of other arms (as with elimination-style algorithms) are achieved by deterministically pruning and meta-pruning. These are only minor parts of our algorithm. Our significant contributions are described below:
> > >
> > > In the stochastic case, after (meta-)pruning the arms, GLUE sets arm indices according to $\psi_k$ (defined in Equation 7) which can be lower than the improved subgaussian factor. In other words, $\psi_k\leq\sigma_k$.  As a result, for suboptimal arms, GLUE's index need NOT be a valid upper confidence bound on the respective mean. This causes the suboptimal arms to be under-explored in comparison to vanilla UCB algorithm, thus leading to improved regret performance. In fact, we outperform Improved UCB both theoretically and empirically; Improved UCB suffers an asymptotic regret of $\sum_{k\in[K_2(0)]} \frac{2\sigma_k^2}{\Delta_k}\geq R_{\infty}^{GLUE}$ where $R_{\infty}^{GLUE}$ is the asymptotic upper bound of GLUE defined in Equation 9. Empirical validation of this behavior is provided in Figure 3.
> > >
> > > In the linear case, R-OFUL exploits the linear structure of the environment to restrict arms after deterministic pruning. This is done by constructing a cone around the most promising arm using the improved subgaussian factors as defined in Equation 3. Futher, the rate of exploration is defined by the worst-case arm in this cone. This leads to additional restrictions that are not achieved by simply clipping the arm indices at the respective mean bounds.

---

### Review · Reviewer_bMxB · 2024-08-08

**Summary Of Contributions:**

The paper considers a version of the stochastic multi-armed bandit and stochastic linear bandit problems where in addition to standard assumptions about the reward's bounds/distribution, the decision-maker is also made aware of some upper and lower bounds on the expected rewards of actions.

The paper contributes an adaptation of OFUL (Abbasi-Yadkori et al., 2011) to incorporate information provided by the bounds in the linear bandit setting, and shows that this can achieve a lower high-probability bound on the worst-case regret than OFUL when the bounds are useful. For the multi-armed bandit setting, an algorithm with features of both UCB1 and KL-UCB is proposed, but which ends up not necessarily behaving optimistically, which again can improve over the regret of traditional algorithms when the mean bounds provide useful capacity to rule out certain actions or explore them at a lower rate.

The work is underpinned by a result showing that knowledge of bounds for the mean of bounded variables can yield sharper sub-Gaussianity results than those achievable using the bounds on the outcome alone, and is motivated by an example in treatment prescribing where an ML/AI tool may have access to less information than a physician, due to the absence of sensitive or otherwise unobserved variables, and thus only know the mean benefit of a treatment up to some best/worse-case bounds.

**Audience:**

Yes

**Broader Impact Concerns:**

There is no statement, but I do not have concerns about direct impacts of this work.

**Claims And Evidence:**

Yes

**Requested Changes:**

1. I'd like to see the introduction re-written to improve accessibility for readers unfamiliar (or rusty on) bandits and confounding/healthcare applications.

2. p1: 'If a log is generated without... alternative': doesn't that depend on what the distribution of times since stroke are in the training/log sample?

2. The figures at times present standard deviation bounds that fall below 0. I think it would be better to plot minima and maxima rather than plus/minus 1 standard deviation, as the mean-1sd falling outside the range suggests a skewed distribution of the regret.

3. p6, I'd say the agent 'aims to minimize' their regret, rather than necessarily does so.

4. The idea of approximating kl-UCB to offer a computational speed-up seems to have been explored elsewhere - I'd recommend referencing and making connections to work such as Lin, Wang, Buccapatnam and Shroff (2018, IJCAI) "UCBoost: A Boosting Approach to Tame Complexity and Optimality for Stochastic Bandits"

5. p10: 'number of played' doesn't seem right grammatically

6. p12: I think asymptotic regret is only really defined within Theorem 6, somewhat implicitly, may it be beneficial for the discussion here to have defined it unambiguously in the main text beforehand?

7. There are some open questions and interesting points for further work identified throughout the paper. I think a concluding Section 7 would be beneficial to pull these together.

**Strengths And Weaknesses:**

Strengths: The problem is interesting and the methods developed are nicely connected to the existing literature and explored to an appropriate extent through a mixture of theoretical and empirical analysis. From what I can tell this theoretical work is conducted accurately, the empirical work shows good performance of the methods proposed and is reproducible. The authors have done a good job in many sections of pre-empting potential queries or points of confusion and justifying the choices made and/or highlighting the benefits of the methods proposed.

Weaknesses: Section 1 is not especially accessible. It starts with quite a technical observation about sub-Gaussian random variables, which I believe is intended as a motivation to the work, but is confusing before defining bandit problems etc. While I appreciate this is a paper that ultimately will be interesting to a technical audience, it would still be good to offer some definitions/introductions to key concepts rather than using the terminology of bandits, regret, etc. and the notion of what confounded of logs are before discussing them. Later in the paper, these concepts are addressed successfully. The other main weakness is in the comparison of theoretical upper bounds to lower bounds for the mean-bounded setting. The paper defers these to further work, and I think for TMLR there is enough here without lower bounds, but it would have been nice to at least see a bit more of a flavour of what the key differences between established lower bounds and those specific to this setting are likely to be, or at least conjectured to be.

---

> ### Author Response · Authors · 2024-09-14
> **Response to Reviewer Comments**
>
> Thank you for your review of our paper and appreciate your comments that will help improve the presentation of our work. We address these in sequence below.
>
> # 1. Section 1 Rewrite:
>
> Following your suggestion, we have restructured Section 1 to improve clarity for a wider population of readers.
>
> # Lower Bounds:
>
> We agree that specifying lower bounds for any online decision making problem is important to judge the efficacy of an upper bound offered by any algorithm.
>
> Asymptotic optimality in the case of linear bandits are achieved by complicated non-optimistic algorithms and are not practically applicable (see Chapter 25 of Bandit Algorithms by Lattimore and Szepesv\'ari for a summary). For this reason, we chose to explore an OFUL-like (or LinUCB-like) algorithm that is computationally light and did not make any claims about optimality. As we discuss in Section 5, for the case of stochastic MABs, in the case of Gaussian and Bernoulli rewards, the OSSB algorithm has been proven to achieve asymptotically optimal performance when the reward distribution is known a priori. In Appendix D, we prove that for the case of Bernoulli rewards, B-KL-UCB matches the lower bound. This is deferred to the appendix as it is not our primary result.
>
> To summarize, the questions of asymptotic optimality in the linear case and for the stochastic setting with general reward distributions remain open. Since our efforts were concentrated in providing efficient arm selection policies that improve over existing work, we have chosen not to comment on the structure of the lower bounds further. We include it as an important and immediate avenue for future work in the new Conclusion section at the end of the paper.
>
> # 3. Comparison to UCBoost:
>
> We have added a reference to this work in Section 5. As you stated, it is relevant in that it aims to provide an efficient version of kl-UCB in the standard bandit setting without significantly compromising regret. Applied to the case of bandits with mean bounds, the natural modification to UCBoost remains an optimistic UCB-style algorithm with its performance being tied to the set of number of quality of semi-distance functions it considers. In contrast, GLUE is a non-optimistic algorithm that potentially explores arms using information from other, seemingly independent, arms.
>
> # 4. Conclusion Section:
>
> Following your (and Reviewer q8Xf's) suggestion, we have included a conclusion section at the end of the paper.
>
> # 5. Negative regret in empirical evaluations:
>
> The definition of regret in the linear bandit case is a difference of {\em mean rewards} of the genie and R-OFUL. To implement this, for each sample path, we track rewards obtained by the respective policies and compute their difference. This leads to the per-instance regret being negative in some rounds. This leads to the negative confidence bars, and in fact a negative minima. The alternative would be to compute the per-instance regret to be the difference between the mean values of the chosen arms at each time, which would lead to the always positive regret trajectory.
>
> While both of these lead to the same average value of regret, we believe that computing regret according to the per-instance rewards rather than per-instance mean rewards is a fairer way to measure regret since the mean rewards are unknown to designed policy a priori. Thus, we have not changed the figures in the revision.
>
> # 6. Distribution of logs in Stroke example:
>
> As long as there exist patients that are administered tPA within 3 hours of occurance of the stroke, the log will always suggest that tPA is superior due to higher average reward (chance of recovery). As the time since onset of stroke is not recorded, the efficacy of the treatment can only be judged through the rewards.
>
> # 7. Text Edits:
>
> Thank you for catching the typos, we have fixed them. We have also added a definition for asymptotic regret before the statement of the Theorem.

---

> > ### Comment · Reviewer_bMxB · 2024-09-16
> >
> > Hi Authors,
> >
> > Thank you for your systematic response, and the revision which I feel does substantially improve the accessibility of the paper. I am making a positive recommendation for the paper.

---

> > > ### Author Response · Authors · 2024-09-16
> > >
> > > Thank you for your feedback on our work.

---

### Review · Reviewer_q8Xf · 2024-08-31

**Summary Of Contributions:**

The authors consider the MAB problem, where additional side information on the rewards distribution is available. This feature makes the mean estimate tighter and improves the method's overall convergence. Based on this approach, the authors propose a Restricted-set OFUL algorithm for the linear bandits setting and a Global Under-Explore algorithm for the stochastic setting. The corresponding regret bounds are analyzed and compared with alternatives that do not consider additional information. The presented approach is evaluated experimentally in the learning from confounded logs, where the target bounds on the mean values appear naturally.

**Audience:**

Yes

**Broader Impact Concerns:**

No concerns about the ethical implications.

**Claims And Evidence:**

Yes

**Requested Changes:**

Some requests are listed inside the weakness list above, the others are presented below

1. Please add the link to the source code (e.g., in GitHub repository) of the developed methods and scripts to reproduce the results of the experiments.
2. Add the Conclusion and Future work sections, where the summary of the presented results and limitations & potential further improvements should be discussed.
3. Please discuss the limitations of the presented approach. In particular, I see two directions:
- what assumptions on the underlying reward distribution should be satisfied? and what happened if they are failed? Do the presented methods robust to such scenarios?
-  is it possible to improve the presented bounds further? Is it possible to derive the lower bound for the mean interval corresponding to the discussed type of the additional information?
4. Please add discussion of the domain-specific problem that can be solved with your approach and what is the interpretation of the additional bound for mean values.

**Strengths And Weaknesses:**

**Strenghts**

1. Detailed description of the contribution and thorough review of the related works
2. Experimental evaluation of the presented methods and description of the scenario where the introduced framework naturally works
3. The regret bounds corresponding to the introduced algorithms are derived and compared with alternatives

**Weaknesses**

1. No Discussion/Conclusion/Future work sections
2. The application problem remains unclear from the description of the confounded logs extracted from the Movielens dataset. This data are typically used in recommender system tasks, do you solve rating prediction problem with your approach? Please provide more details on the domain problem that is converted to linear bandits and stochastic MAB problems.
3. Missing discussion of the limitations for the suggested methods and potential improvements for the developed framework

---

> ### Author Response · Authors · 2024-09-14
> **Response to Reviewer Comments**
>
> We begin by thanking the reviewer for their comments that will improve the quality of the paper. All the code used to generate the results in the paper have already been included as part of the supplementary material. These can be found in the zip file associated with the submission.
>
> We address the questions with clarifications below:
>
> # 1. Discussions, Conclusions and Future Work sections:
>
> We provide comparisons with existing work, discussions and avenues for future work for the bandit problem at the end of their respective sections. These can be found in Section 4.3 and 5.3 respectively.
>
> Following your (and Reviewer bMxB's) suggestion, we have also also included a Conclusion section that summarizes our contributions and future directions at the end of the main paper to improve the presentation.
>
> # 2. Confounded Logs with Movielens:
>
> The application problem we consider is an online variant of the recommendation system task that you mention. With reference to our stochastic MAB setting with Movielens, under each visible context (the occupation feature), the agent is tasked with learning the best movie to recommend in an online manner. The confounded logs can be used to infer bounds on the mean rating of each movie using our results in Theorem 7 and help accelerate the learning process through GLUE (Algorithm 2).
>
> In the linear case, we present synthetic experiments that follow the setting described in Section 6.2. In this case, the oracle has access to the full latent vector $(\theta^*_z, T_u)$ at each time, while the agent only has access to $\theta^*_z$, but must play arms in order to minimize regret with respect to the unknown quantity $\theta^*_u$ (the average of the vector $T_u$). Mapping this to a recommender system example, $T_u$ represents the personal preference of each user, while $\theta^*_u$ is the average preference of the population. Invariant effects, such as the likelihood of movie watching among people in a given postal code, are captured by $\theta^*_z$.
>
> We have modified the text in Section 6.3 to better explain the recommendation systems task. Further, we have also added interpretations of the linear bandit problem in our healthcare example (introduced in the Introduction) in Section 6.2.
>
> # 3. Limitations and Potential Improvements:
>
> The key assumption in our work is that of bounded rewards. This assumption allowed us to infer and use improved subgaussian factors in the linear and MAB problems. This assumption is widely used in linear bandit literature, however, stochastic MAB algorithms like UCB and KL-UCB work without this assumption. While our methods remain applicable in cases where one can derive such sharp subgaussian factors for arbitrary reward distributions, they do not readily extend to unbounded reward settings.
>
> As mentioned in (the existing) Section 4.3, one immediate avenue for future work would be to improve our regret bounds in the linear bandit setting. It is well known that for standard linear bandit problems, the optimal policy is a non-optimistic one. However, these involve expensive computations at each step and thus, we chose to study a light-weight OFUL-like algorithm due to its practicality. In the stochastic MAB setting, as we describe in Section 5, OSSB is provably optimal for Bernoulli and Gaussian reward settings (and reduces to B-KL-UCB in the former case). The question of optimality for general reward distributions remains open.
>
> Another possible thread would be to treat the case with probabilistic knowledge of the mean bounds. We discuss this in (the existing) section 5.3 and also provide Corollary 6.1 which uses our methods as a baseline.
>
> In the (new) Conclusion section at the end of the paper, we have recalled the lower bound and probabilistic mean bounds as avenues for future work and added the case of unbounded rewards as a limitation.
>
> # 4. Domain-specific Problem:
>
> We discuss two domain specific tasks in the paper. In the Introduction, we present a healthcare use-case to motivate the use of online learning in the presence of confounded logs. We treat the recommendation system problem with the Movielens dataset in Section 6.3 (please see point 2 above for more details).
>
> The former was discussed further in Section 6.1.1 where we interpret our assumptions in this setting. As mentioned in point 2 of the response, we have also added such interpretations for this use case in Section 6.2. Further, Section 6.3 has also been modified to better explain the recommendation system setting.

---

> > ### Comment · Reviewer_q8Xf · 2024-09-15
> >
> > Dear authors,
> >
> > Thanks for the detailed response to my comments and the proper revision of the manuscript!

---

> > > ### Author Response · Authors · 2024-09-16
> > >
> > > Thank you for your feedback on our work.

---

### Author Response · Authors · 2024-09-14
**Revision**

We thank all the reviewers for their comments and feedback on our paper. Addressing these, we have the modified the paper and the PDF attached to this submission has been updated. We have also updated the full paper in the supplementary zip.

The primary modifications to the paper are a restructured Introduction section, and a new Conclusion section at the end of the paper. Further Sections 6.2 and 6.3 now include further clarification on the practical use cases. Minor textual edits (following recommendations by Reviewer bMxB) have also been incorporated.

---

### Author Response · Authors · 2024-10-27
**Camera-ready**

Once again, we thank all the reviewers as well as the action editor for their time and feedback throughout the review process. We have updated the submission with the deanonymized camera-ready version of the paper.

---

### Decision · Action_Editor_sfrk · 2024-10-10

**Recommendation:** Accept as is

**Comment:**

The reviewers think that the manuscript suggests a novel approach for incorporating side information in the update rule in linear and MAB setup. Although there were concerns about technical novelty and the experimental evaluation of the proposed approach is perhaps limited, they overall agree that this is a solid contribution to TMLR.

**Audience:**

Yes. The work is grounded in important applications such as medical treatment and recommendation, where confounding naturally happens.

**Claims And Evidence:**

Yes. The authors' rebuttal addressed the reviewers' technical concerns well.